# Closing the Loop: Universal Repository Representation with RPG-Encoder

Jane Luo [1‡*]   Chengyu Yin [1‡*]   Xin Zhang [1*†]   Qingtao Li [1]   Steven Liu [1‡]   Yiming Huang [2]
Jie Wu [3‡]   Hao Liu [1‡]   Yangyu Huang [1]   Yu Kang [1]   Fangkai Yang [1]   Ying Xin [1]   Scarlett Li [1]

## Abstract

Current repository agents encounter a reasoning disconnect due to fragmented representations, as existing methods rely on isolated API documentation or dependency graphs that lack semantic depth. We consider repository comprehension and generation to be inverse processes within a unified cycle: generation expands intent into implementation, while comprehension compresses implementation back into intent. To address this, we propose RPG-Encoder, a framework that generalizes the Repository Planning Graph (RPG) from a static generative blueprint into a unified, high-fidelity representation. RPG-Encoder closes the reasoning loop through three mechanisms: (1) Encoding raw code into the RPG that combines lifted semantic features with code dependencies; (2) Evolving the topology incrementally to decouple maintenance costs from repository scale, reducing overhead by 95.7%; and (3) Operating as a unified interface for structure-aware navigation. In evaluations, RPG-Encoder establishes state-of-the-art repository understanding on SWE-bench Verified with 93.7% Acc@5 and exceeds the best baseline by over 10% on SWE-bench Live. These results highlight our superior fine-grained localization accuracy in complex codebases. Furthermore, it achieves 98.5% reconstruction coverage on RepoCraft, confirming RPG's high-fidelity capacity to mirror the original codebase and closing the loop between intent and implementation. Our code and data are available at https://github.com/microsoft/RPG-ZeroRepo.

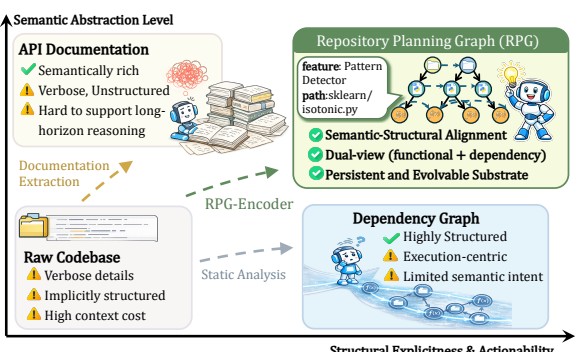

*Figure 1.* Comparison of code representations regarding semantic abstraction and structural explicitness. Unlike approaches limited to a single dimension, RPG achieves dual-view alignment, combining semantic richness with structural actionability.

## 1. Introduction

Repository-level software engineering relies on agents navigating complex dependencies and reasoning about high-level architectural intent (Wang et al., 2025a; Zhao et al., 2025). However, as illustrated in Figure 1, existing approaches suffer from a reasoning gap due to fragmented representations: **API Documentation** focuses on semantic intent (Luo et al., 2024; Chen et al., 2025a) but lacks global navigability, forcing models to infer architectural connectivity (Chen et al., 2025b; Jain et al., 2025). Conversely, **Dependency Graph** captures structural logic (Ouyang et al., 2024; Ma et al., 2024) but provide limited semantic information (Borowski et al., 2024; Cheng et al., 2024), leaving agents to follow execution paths without reflecting the underlying rationale (Jiang et al., 2025). Furthermore, maintaining consistency incurs prohibitive overhead: documentation is prone to semantic drift (Tan et al., 2024), while static graphs capture syntactic updates but often overlook logical implications (Gröninger et al., 2025).

We observe that this reasoning disconnect is not merely a failure of individual tools, but a systemic consequence of treating repository understanding as an isolated, unidirectional task. Fundamentally, this occurs because current approaches ignore the inherent symmetry of software engineering. We argue that repository comprehension and generation constitute inverse pathways within a unified reasoning

---

[*]Equal contribution   [†]Corresponding author   [‡]Work done during internships at Microsoft  [1]Microsoft [2]University of California San Diego [3]Tsinghua University. Correspondence to: Xin Zhang <xinzhang3@microsoft.com>.

*Proceedings of the 43rd International Conference on Machine Learning*, Seoul, South Korea. PMLR 306, 2026. Copyright 2026 by the author(s).

cycle: generation expands sparse intent into detailed code, whereas comprehension must compress noisy implementation back into high-level intent. Consequently, bridging this gap requires a **unified Intermediate Representation** that fuses the semantic density of documentation with the topological rigor of dependency graphs. The Repository Planning Graph (RPG) (Luo et al., 2025) emerges as a suitable representation for this unification. Having served as a generative blueprint for intent-to-code, it possesses the dual-view structure needed for the inverse code-to-intent journey. This motivates our fundamental inquiry: *Can the RPG be generalized to serve as a unified, high-fidelity representation for existing repositories, thereby closing the loop?*

To realize this vision, we propose RPG-Encoder, a framework that transforms the RPG from a static generative blueprint into a dynamic, bidirectional representation. We implement this through three cohesive mechanisms: (1) **Encoding:** We introduce a semantic lifting protocol that projects code into the RPG. Nodes combine functional descriptions with code metadata, while edges encode hierarchy and static dependencies, yielding an interpretable and verifiable representation. (2) **Evolution:** We design an incremental mechanism that parses commit diffs to update the RPG. This keeps semantics synchronized with implementation without re-generation. (3) **Operation:** We establish the RPG as a unified interface for structure-aware reasoning. It serves as a topological map, enabling traversal between high-level intent and low-level execution logic.

To evaluate the extracted RPG, we evaluate two dimensions: navigational utility and representational fidelity. (1) In **Repository Understanding**, RPG-Encoder with Claude-4.5-Sonnet (Anthropic, 2025b) demonstrates superior function-level performance, achieving 93.7% Acc@5 on SWE-bench Verified (OpenAI, 2024) and exceeding the best baseline by over 10% on SWE-bench Live (Zhang et al., 2025). This confirms that coupling semantic features with topology significantly strengthens fine-grained localization. (2) In **Repository Reconstruction**, RPG-Encoder outperforms API documentation by providing an explicitly ordered blueprint. Guided by topological constraints, RPG-Encoder reconstructs repositories with 98.5% coverage (+24.3% over baselines) and 86.0% pass rate on RepoCraft (Luo et al., 2025). In contrast, documentation lacks structural guidance and recovers only ~17% of the code volume, proving that RPG serves as a structured representation that effectively preserves complete repository semantics. Analysis confirms that semantic features are essential for effective exploration, and our incremental strategy reduces maintenance costs by 95.7% without semantic drift.

Our contributions are summarized as follows:

- We generalize the Repository Planning Graph (RPG) into a unified representation that closes the loop between comprehension and generation, theoretically grounding repository reasoning as a unified reasoning cycle where semantic intent and structural dependencies are bidirectionally linked.

- We introduce RPG-Encoder, a framework that implements a semantic lifting protocol to recover high-level intent from code and supports sustainable evolution via differential updates, decoupling maintenance costs from repository scale.

- We validate RPG-Encoder on dual tasks: establishing SOTA performance in repository understanding to demonstrate superior navigational utility, and achieving 98.5% coverage in repository reconstruction to verify its high-fidelity representational capacity.

## 2. Related Work

**Repository Generation.** Repository generation has progressed from file-level completion (Wang et al., 2025b; Li et al., 2023) to autonomous agentic workflows. Multi-agent frameworks (Hong et al., 2024) and paper-to-code systems (Seo et al., 2025; Lin et al., 2025) orchestrate complex implementations, while terminal interfaces (Anthropic, 2025a; Wang et al., 2024) enable execution. RPG (Luo et al., 2025) further grounds generation with structured planning graphs and explicit dependencies.

**Repository Understanding.** Recent work focuses on localization, including iterative retrieval (Zhang et al., 2023; Xia et al., 2024) and graph-guided navigation (Ouyang et al., 2024; Liu et al., 2025). Recent agents combine planning (Yu et al., 2025) and dependency search (Jiang et al., 2025) within executable loops (OpenAI, 2024; Yang et al., 2024), and incorporate knowledge graphs (Wang et al., 2025a; YANG et al., 2025) to improve context retention.

## 3. Method

To establish RPG as a unified and high-fidelity Intermediate Representation, we introduce the RPG-Encoder. By mapping implementation back to semantic space (Code $\rightarrow$ RPG), it completes the representation loop. As illustrated in Figure 2, our methodology comprises: (1) Encoding for RPG extraction; (2) Evolution for incremental maintenance; and (3) Operation as a unified reasoning substrate.

### 3.1. RPG Encoding: Extracting RPG from Codebases

To transform a raw codebase into an actionable substrate, we model extraction as a pipeline that converts implementation details into a compact, structured semantic index for high-level reasoning. This process reconstructs the system topology in three phases. More details are in Appendix A.1.

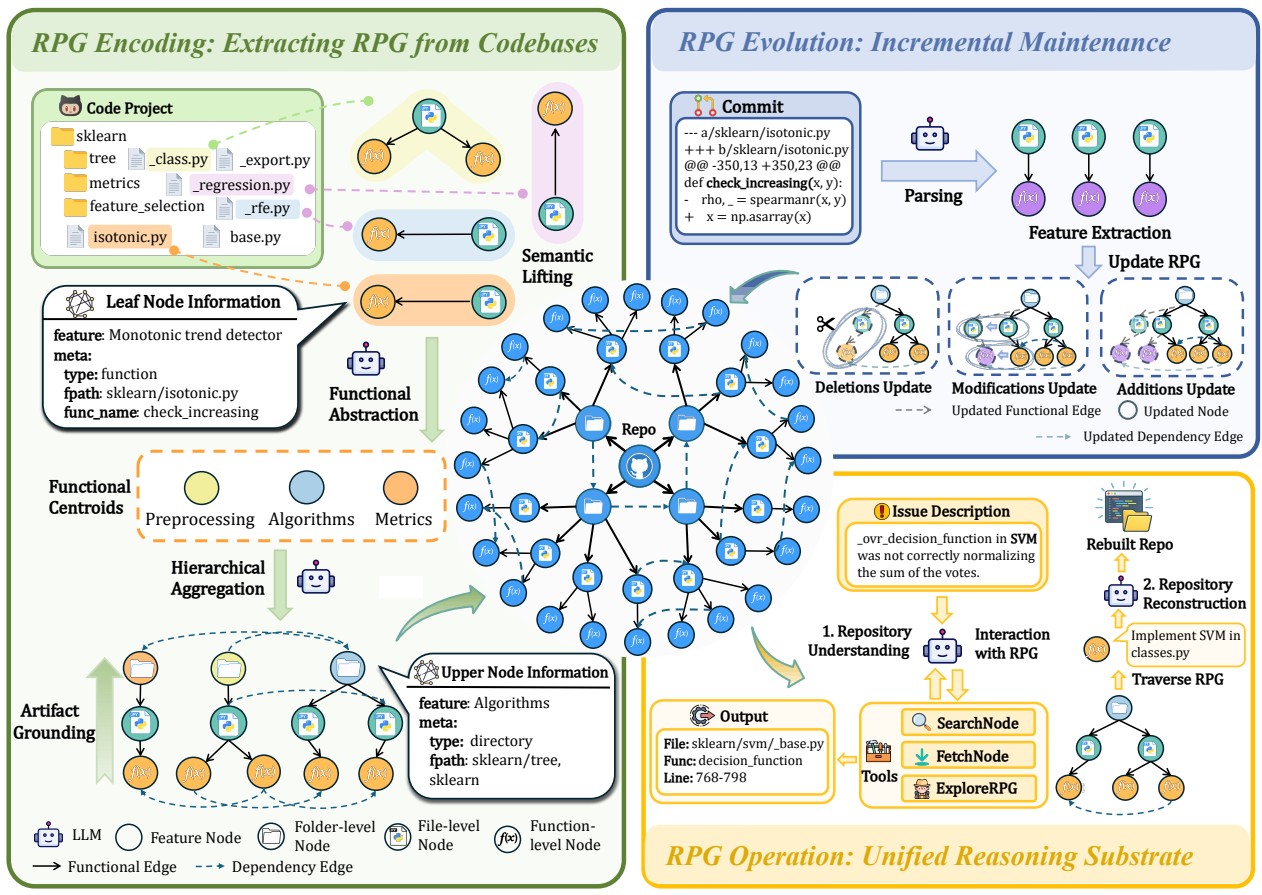

Figure 2. Overview of the RPG-Encoder. The pipeline bridges Code and RPG via three stages: **Encoding** lifts code into a semantic topology; **Evolution** handles incremental updates via commits; and **Operation** provides a unified interface for agentic reasoning.

**RPG Structure** Refining prior definitions ([Luo et al., 2025](#)), we define RPG as a hierarchical, dual-view graph $\mathcal{G} = (\mathcal{V}, \mathcal{E})$. The node set $\mathcal{V} = \mathcal{V}_H \cup \mathcal{V}_L$ distinguishes High-level Nodes representing architectural directories from Low-level Nodes comprising atomic implementations such as files, classes, and functions. Each node $v = (f, \mathbf{m}) \in \mathcal{V}$ pairs a semantic feature $f$ describing functionality (e.g., *handles authentication*) with structural metadata $\mathbf{m}$ encoding code entity attributes like type and path. The edge set $\mathcal{E}$ integrates two perspectives: (1) Functional edges $\mathcal{E}_{\text{feature}}$ establishing teleological hierarchy; and (2) Dependency edges $\mathcal{E}_{\text{dep}}$ mapping logical interactions including imports and calls. This duality enables the agent to perceive the repository as both a functional and executable network.

**Phase 1: Semantic Lifting** To bridge the granularity mismatch between verbose implementation and functional intent, the extraction process first lifts the codebase into a discrete registry of Low-level Nodes ($\mathcal{V}_L$). For each file, the system extracts semantic features $f$ for individual functions and classes, mapping them to behavioral signatures while retaining their code-level attributes as metadata $\mathbf{m}$. Subse-

quently, these fine-grained features are synthesized into a holistic summary representing the file's overall functionality. This summarization process naturally establishes functional edges $\mathcal{E}_{\text{feature}}$ between the file-level node and its constituent function-level node. This phase concludes by producing a semantically grounded implementation index, serving as a robust representation for higher-level reasoning.

**Phase 2: Semantic Structure Reorganization** Physical folder-file organization is often dictated by technical constraints rather than functional boundaries, inducing structural entanglement. To mitigate this, we construct the High-level Node set $\mathcal{V}_H$ by recovering the latent functional topology from implementation units ($\mathcal{V}_L$). (1) **Functional Abstraction:** To ensure the global repository state fits within the LLM context window, we perform a granularity-based input compression. Instead of raw implementation, the LLM only consumes the concise semantic features $f$ of file-level nodes, excluding function-level details. This condensed view allows the model to analyze the complete repository-wide semantic manifold to induce abstract functional centroids (e.g., *Data Preprocessing*) that define the root pillars

of the hierarchy. (2) **Hierarchical Aggregation:** We recursively link nodes from $\mathcal{V}_L$ to these centroids. To ensure structural stability, each node's placement is determined by a semantic compatibility check: the LLM evaluates the fit between a node's $f$ and the centroid's definition, instantiating intermediate nodes (e.g., routing *StandardScaler* via *Normalization* to *Preprocessing*) to bridge the hierarchy when a direct link lacks sufficient granularity. Together, these nodes constitute the High-level Node set $\mathcal{V}_H$, establishing explicit parent-child functional edges. This results in a complete functional graph where each $v \in \mathcal{V}_H$ possesses a semantic feature $f$ but lacks the structural metadata $\mathbf{m}$ required to link it to physical code entities.

**Phase 3: Artifact Grounding** To transform the abstract hierarchy into a substrate, this phase anchors the functional manifold to physical artifacts and execution logic. We first populate the missing metadata $\mathbf{m}$ for nodes in $\mathcal{V}_H$ through bottom-up propagation, utilizing a Lowest Common Ancestor (LCA) mechanism (detailed in Appendix A.1.3) to compute the minimal directory scope shared by each cluster's descendants. This mapping ensures that abstract features such as *Data Preprocessing* are tied to code paths like `sklearn/preprocessing`. Subsequently, to transition from a semantic hierarchy to an implementation map, we inject dependency edges $\mathcal{E}_{\text{dep}}$ (e.g., imports, calls) via AST analysis. This integration completes the RPG, yielding a unified representation that enables traceability between high-level functional intent and the executable code.

### 3.2. RPG Evolution: Incremental Maintenance

To reduce the cost of full re-generation, we maintain $\mathcal{G}$ incrementally and reserve global reconstruction for major refactoring. For routine updates, we perform online graph editing to keep the RPG synchronized, as illustrated in Figure 2 (top-right) and detailed in Appendix A.2.

**Commit-Level Feature Extraction** We parse raw commit data to extract semantic features strictly for affected code fragments, avoiding full reprocessing. This yields a set of discrete Feature Nodes representing the delta state, which serves as the direct input for graph operations.

**RPG Updates** Based on the diff type, we execute three atomic update protocols to maintain the RPG structure: (1) **Deletions**: We remove nodes for deleted files or functions and recursively prune empty parent categories in $\mathcal{V}_H$ to maintain hierarchical integrity. (2) **Modifications**: We regenerate the semantic description $f$ for modified entities. To avoid unnecessary structural instability, a node's position is updated only if the LLM detects a functional intent shift that violates its current parent's semantic scope (e.g., a utility function evolving into a core algorithm). This check serves

as a semantic threshold to prevent minor implementation changes from triggering costly structural migrations. (3) **Additions**: We create nodes for new entities and insert them into the hierarchy by matching their semantics against existing functional centroids. Finally, we perform a localized dependency update, re-parsing affected ASTs to refresh $\mathcal{E}_{\text{dep}}$ and align connectivity with the current execution flow.

### 3.3. RPG Operation: Unified Reasoning Substrate

We deploy RPG as a Unified Representation providing a queryable index of the codebase. Structurally, it functions as a heterogeneous graph where Functional and Dependency Views are partitioned by edge types ($\mathcal{E}_{\text{feature}}$ and $\mathcal{E}_{\text{dep}}$) but share a unified node set, enabling seamless context switching during retrieval. More details are in Appendix A.3

**Unified Agentic Tool** We define three core tools to operate on the RPG's nodes and edges:

- **SearchNode**: Performs global node-level retrieval by matching intent against semantic features $f$ or filtering metadata $\mathbf{m}$, allowing the agent to precisely localize entry points across both views.

- **FetchNode**: Executes node-level data retrieval. Given $v$, it extracts the attribute tuple $(f, \mathbf{m})$ and raw source code to provide the ground truth for inspection.

- **ExploreRPG**: Facilitates cross-view traversal along edges $\mathcal{E}$. While $\mathcal{E}_{\text{dep}}$ is strictly constructed via static AST analysis, its integration with the semantic hierarchy in $\mathcal{V}_H$ provides a robust topological skeleton that guides the agent through complex execution flows without the noise of unstructured search.

This toolset enables multi-dimensional navigation by integrating functional intent with physical implementation, facilitating precise context discovery through semantic and dependency structures.

**Efficient Structured Representation** RPG reduces information overload by representing the repository as a substrate with two roles: (1) Knowledge Source: RPG stores feature descriptions and metadata for each node, capturing *what* the code does without parsing implementations. (2) Process Encoder: RPG induces a topological order via functional edges ($\mathcal{E}_{\text{feature}}$) and dependency edges ($\mathcal{E}_{\text{dep}}$), exposing causality and hierarchy essential for architectural comprehension.

## 4. Experiments Setup

We evaluate RPG on two tasks to assess its semantic grasp and structural completeness: (1) Repository Understanding, testing navigation and localization capabilities; and

(2) Repository Reconstruction, verifying the fidelity and losslessness of the encoded information.

## 4.1. Repository Understanding

We assess RPG as a navigational substrate through rigorous localization tasks. More details are in Appendix B.1.

**Benchmark.** We evaluate on two benchmarks: SWE-bench Verified (OpenAI, 2024), a human-validated subset ensuring solvability with 500 examples from 12 repositories; and SWE-bench-Live Lite (Zhang et al., 2025), mitigating contamination using recent issues, comprising 300 examples across 70 repositories.

**Baselines.** We compare against baselines leveraging diverse structural priors: Agentless (Xia et al., 2024) operates via hierarchical text-based narrowing without graph priors; LocAgent (Chen et al., 2025c) leverages explicit dependency graphs for guided traversal; CoSIL (Jiang et al., 2025) performs iterative search over static code structures; and OrcaLoca (Yu et al., 2025) integrates dynamic execution signals with agentic planning.

**Evaluation Metrics.** We adopt standard metrics: Acc@k ($k \in \{1, 5\}$) checks if a ground-truth target is in top-$k$ predictions (Jiang et al., 2025); and Precision/Recall quantify overlap. Given predicted set $P$ and ground-truth $G$, we define Precision $= |P \cap G|/|P|$ and Recall $= |P \cap G|/|G|$.

**Implementation Details.** We use GPT-4o (OpenAI, 2024) to parse and incrementally update the RPG. Backbone models include o3-mini (OpenAI, 2025d), GPT-4o (OpenAI, 2024), GPT-4.1 (OpenAI, 2025a), GPT-5 (OpenAI, 2025b), DeepSeek-V3.1 (Liu et al., 2024), and Claude-4.5-sonnet (Anthropic, 2025b). RPG-Encoder operates with a 40-step limit. Baselines follow configurations (detailed in Appendix B.1.1). All runs are averaged over 3 times.

## 4.2. Repository Reconstruction

We use reconstruction to verify lossless, topologically ordered RPG information. Details are in Appendix B.2.

**Benchmark.** We adapt RepoCraft (Luo et al., 2025) for controlled reconstruction, aiming to rebuild target repositories (e.g., `Requests`) with ground-truth functionality. To isolate representational fidelity, we compare Official API Documentation with RPG. We focus on representation sources rather than search-based agents (e.g., LocAgent (Chen et al., 2025c)), since reconstruction requires a comprehensive blueprint instead of iterative localization.

**Baselines.** We configure ZeroRepo (Luo et al., 2025) in two modes: (1) **ZeroRepo-Doc (Baseline):** The agent

references API documentation, autonomously managing progress and objectives via Test-Driven Development. (2) **ZeroRepo-RPG (Ours):** We utilize the extracted RPG for direct repository generation, where it serves as the exclusive knowledge source and scheduler. Nodes are processed in topological order, batching semantically similar nodes to accelerate inference. More details are in Appendix B.2.2.

**Evaluation Metrics.** Following RepoCraft, we report: (1) Coverage, the proportion of implemented functional categories; (2) Accuracy (Pass / Vote), unit-test pass accuracy and vote-based check accuracy; and (3) Code Statistics (#Files, nLOC, Code Tokens) to measure structural similarity and recovered code volume.

**Implementation Details.** We employ GPT-4o (OpenAI, 2024) for RPG extraction and evaluate reconstruction using GPT-5-mini (OpenAI, 2025c) and GPT-4.1 (OpenAI, 2025a). Following RepoCraft, we also use o3-mini (OpenAI, 2025d) for automated evaluation. ZeroRepo-Doc runs without a hard turn limit and stops when the agent judges the documentation to be fully implemented. ZeroRepo-RPG is bounded by the graph and terminates once all RPG-derived nodes are executed. More details are in Appendix B.2.4.

## 5. Main Result

**RPG Enhances Fine-Grained Repository Understanding.** Table 1 demonstrates that RPG consistently improves file-level and function-level localization. On SWE-bench Verified, RPG-Encoder with Claude-4.5 achieves 93.7% Acc@5 on function level, surpassing the best baseline (OrcaLoca) by 14.4 points, while simultaneously improving Precision by 6.9% and Recall by 10.7%. Furthermore, on SWE-bench Live, RPG-Encoder with GPT-5 elevates performance to 87.8% Acc@5 on function level, outperforming CoSIL by 11.6 points. These results confirm that coupling semantic features with topological constraints enables agents to map high-level intent to specific implementation units. Crucially, this dual-view structure filters irrelevant noise while ensuring comprehensive coverage of target functionalities.

**RPG Functioning as a Complete Representational Substrate.** Table 2 demonstrates RPG's superior fidelity in reproducing complex repository structures. With GPT-5-mini, RPG-Encoder attains 98.5% Coverage and an 86.0% Pass Rate, exceeding the documentation-based baseline by over 33 points. Regarding code scale, the baseline generates severely fragmented outputs, capturing only ~17% of the original volume due to a lack of structural guidance. In contrast, RPG-Encoder reconstructs 550k tokens, a scale comparable to the gold project written by human. This high fidelity proves that RPG serves as a sufficient substrate to ground architectural intent within a valid structural topol-

*Table 1.* Comprehensive localization results on SWE-bench Verified and SWE-bench Live Lite across File and Function levels. Acc@k: Accuracy@k. Pre/Rec: Precision/Recall. **Bold** indicates the best result, and Underline indicates the second best.

| Method | SWE-bench Verified | | | | | | | | SWE-bench Live | | | | | | | |
|---|---|---|---|---|---|---|---|---|---|---|---|---|---|---|---|---|
| | File-level | | | | Function-level | | | | File-level | | | | Function-level | | | |
| | Acc@1 | Acc@5 | Pre | Rec | Acc@1 | Acc@5 | Pre | Rec | Acc@1 | Acc@5 | Pre | Rec | Acc@1 | Acc@5 | Pre | Rec |
| **Model: o3-mini** | | | | | | | | | | | | | | | | |
| Agentless | 67.1 | 88.1 | 67.0 | 64.7 | 34.7 | 60.3 | 39.4 | 33.2 | 54.2 | 78.5 | 55.6 | 47.7 | 28.8 | 54.2 | 39.3 | 25.6 |
| OrcaLoca | 67.5 | 71.9 | 68.3 | 64.0 | 46.3 | 52.9 | 48.3 | 41.5 | 35.4 | 38.0 | 36.2 | 27.6 | 23.1 | 26.1 | 25.3 | 15.6 |
| LocAgent | 62.8 | 77.2 | 64.7 | 61.4 | 32.1 | 40.5 | 33.9 | 28.9 | 47.6 | 59.4 | 49.7 | 41.2 | 23.8 | 31.0 | 26.6 | 17.7 |
| CoSIL | 66.5 | 85.7 | 66.2 | 63.6 | 52.2 | 73.3 | 54.7 | 47.1 | 60.9 | 80.8 | 66.1 | 54.8 | 43.8 | 65.1 | 51.4 | 35.6 |
| **Repo-Enc** | **78.3** | **91.2** | **80.7** | **76.8** | **58.5** | **77.8** | **62.9** | **55.1** | **73.7** | **88.2** | **77.5** | **64.5** | **56.5** | **75.6** | **64.7** | **46.9** |
| $\Delta_{best}$ | +10.8 | +3.1 | +12.4 | +12.1 | +6.3 | +4.5 | +8.2 | +8.0 | +12.8 | +7.4 | +11.4 | +9.7 | +12.7 | +10.5 | +13.3 | +11.3 |
| **Model: GPT-4o** | | | | | | | | | | | | | | | | |
| Agentless | 63.0 | 86.1 | 63.1 | 61.1 | 31.4 | 58.8 | 34.7 | 29.3 | 56.1 | 78.8 | 57.1 | 48.3 | 30.6 | 57.4 | 41.4 | 26.4 |
| OrcaLoca | 64.3 | 69.3 | 65.0 | 61.4 | 39.8 | 53.3 | 42.5 | 36.7 | 42.5 | 47.6 | 45.0 | 34.0 | 28.2 | 37.0 | 32.5 | 21.1 |
| LocAgent | 71.9 | 87.9 | 73.4 | 69.3 | 40.1 | 67.4 | 44.8 | 38.1 | 62.5 | 80.0 | 66.8 | 54.2 | 35.7 | 56.4 | 44.5 | 29.9 |
| CoSIL | 64.9 | 84.4 | 65.0 | 62.2 | 43.2 | 66.2 | 48.2 | 40.1 | 60.1 | 77.0 | 63.7 | 50.7 | 41.2 | 61.6 | 49.1 | 29.4 |
| **Repo-Enc** | **74.5** | **89.6** | **77.0** | **72.7** | **53.1** | **76.7** | **57.9** | **49.5** | **69.2** | **83.5** | **73.2** | **60.3** | **50.5** | **69.4** | **59.4** | **41.8** |
| $\Delta_{best}$ | +2.6 | +1.7 | +3.6 | +3.4 | +9.9 | +9.3 | +9.7 | +9.4 | +6.7 | +3.5 | +6.4 | +6.1 | +9.3 | +7.8 | +10.3 | +11.9 |
| **Model: GPT-4.1** | | | | | | | | | | | | | | | | |
| Agentless | 65.2 | 90.8 | 65.7 | 63.5 | 29.3 | 49.0 | 32.7 | 26.4 | 62.0 | 85.5 | 63.0 | 54.5 | 35.1 | 59.4 | 46.0 | 25.4 |
| OrcaLoca | 75.2 | 80.0 | 76.5 | 71.3 | 55.2 | 66.7 | 59.0 | 50.1 | 56.2 | 59.6 | 57.1 | 44.2 | 42.0 | 50.5 | 46.2 | 29.1 |
| LocAgent | 79.5 | 90.9 | 80.8 | 77.2 | 32.3 | 65.6 | 36.7 | 31.2 | 74.7 | 87.9 | 76.8 | 66.1 | 43.4 | 68.7 | 52.5 | 38.7 |
| CoSIL | 69.8 | 90.6 | 70.7 | 67.6 | 51.8 | 74.5 | 55.3 | 47.0 | 62.3 | 84.7 | 67.3 | 55.6 | 48.8 | 72.2 | 58.3 | 41.2 |
| **Repo-Enc** | **82.6** | **93.2** | **83.6** | **79.3** | **68.7** | **83.4** | **71.0** | **62.4** | **78.0** | **90.5** | **81.4** | **69.0** | **64.7** | **81.9** | **72.1** | **52.6** |
| $\Delta_{best}$ | +3.1 | +2.3 | +2.8 | +2.1 | +13.5 | +8.9 | +12.0 | +12.3 | +3.3 | +2.6 | +4.6 | +2.9 | +15.9 | +9.7 | +13.8 | +11.4 |
| **Model: GPT-5** | | | | | | | | | | | | | | | | |
| Agentless | 78.7 | 95.9 | 78.3 | 76.2 | 45.1 | 68.1 | 47.3 | 41.3 | 64.5 | 87.4 | 65.1 | 57.4 | 38.8 | 64.6 | 49.7 | 31.6 |
| OrcaLoca | 87.9 | 94.4 | 88.4 | 84.4 | 74.6 | 86.3 | 78.2 | 67.5 | 73.8 | 81.9 | 77.1 | 62.2 | 59.2 | 72.6 | 68.4 | 45.6 |
| LocAgent | 88.2 | 96.7 | 88.4 | 86.7 | 50.9 | 80.3 | 55.9 | 49.7 | 79.7 | 93.0 | 81.4 | 74.2 | 48.0 | 68.7 | 56.6 | 40.5 |
| CoSIL | 82.8 | 95.7 | 82.3 | 80.2 | 68.3 | 81.8 | 68.9 | 62.3 | 69.8 | 89.3 | 72.9 | 62.2 | 55.2 | 76.2 | 62.3 | 46.5 |
| **Repo-Enc** | **91.9** | **97.7** | **91.1** | **89.1** | **83.4** | **93.6** | **84.5** | **76.9** | **82.1** | **94.4** | **85.4** | **76.2** | **71.9** | **87.8** | **78.1** | **61.1** |
| $\Delta_{best}$ | +3.7 | +1.0 | +2.5 | +2.4 | +7.3 | +7.4 | +5.4 | +8.3 | +2.4 | +1.4 | +4.0 | +2.0 | +12.3 | +11.6 | +9.5 | +14.5 |
| **Model: Claude-4.5-Sonnet** | | | | | | | | | | | | | | | | |
| Agentless | 76.6 | 96.5 | 76.9 | 74.4 | 31.7 | 34.6 | 32.0 | 27.1 | 63.8 | 89.7 | 66.1 | 58.0 | 41.4 | 72.4 | 55.3 | 35.9 |
| OrcaLoca | 87.2 | 89.6 | 87.5 | 82.2 | 74.5 | 79.3 | 76.5 | 65.1 | 74.7 | 78.3 | 76.2 | 61.5 | 65.1 | 69.4 | 67.8 | 46.1 |
| LocAgent | 71.4 | 76.6 | 72.7 | 70.2 | 49.3 | 57.8 | 51.5 | 44.9 | 58.7 | 69.0 | 61.6 | 54.7 | 47.3 | 60.5 | 52.6 | 39.3 |
| CoSIL | 75.5 | 96.1 | 75.9 | 73.7 | 57.5 | 78.7 | 60.7 | 52.9 | 64.5 | 88.3 | 69.4 | 57.5 | 51.1 | 74.9 | 60.1 | 39.6 |
| **Repo-Enc** | **90.5** | **97.6** | **91.8** | **88.6** | **79.8** | **93.7** | **83.4** | **75.8** | **82.0** | **93.9** | **85.6** | **75.8** | **74.8** | **90.4** | **80.7** | **63.3** |
| $\Delta_{best}$ | +3.3 | +1.1 | +4.3 | +6.4 | +5.3 | +14.4 | +6.9 | +10.7 | +7.3 | +4.2 | +9.4 | +14.3 | +9.7 | +15.5 | +12.9 | +17.2 |

ogy, guiding the agent to expand the blueprint into concrete implementation unlike linear API documentation.

# 6. Ablation

**Experimental Setup.** To isolate semantic and topological contributions, we run two ablations (detailed in Appendix D). For Reconstruction (RepoCraft), we progressively strip node metadata bottom-up while retaining semantic features to evaluate representational fidelity. For Understanding (SWE-bench Live), we remove structural metadata **m** to assess navigational efficacy under RPG-Encoder.

**Semantics and Topology are Mutually Reinforcing.** Table 3 delineates the distinct functional contributions of graph components. Semantic Features provide essential semantic grounding for fine-grained localization; their re-

moval causes the sharpest decline in Function-level Acc@1 ($50.5\% \rightarrow 43.1\%$ on GPT-4o), indicating that abstract summaries are indispensable for aligning natural language intent with concrete implementations. Dependencies establish structural connectivity; severing these edges disrupts execution tracing, significantly degrading File-level retrieval. The Full RPG integrates these layers to maximize context discovery, consistently outperforming all ablated variants.

**Hierarchical Constraints Ensure Structural Fidelity.** Table 4 indicates that the multi-level topology of RPG is essential for preserving repository modularity. Removing file and function metadata (*w/o File & Function*) results in a notable loss of structure: the number of files decreases from 256 to 157, and code volume drops by approximately 200,000 tokens. This suggests that without explicit topolog-

*Table 2.* Main results on repository reconstruction tasks in RepoCraft. Gold Projects represent statistics of the original human-written repositories, as reported in prior work (Luo et al., 2025). More results are provided in Appendix C.1.

| Framework | Backbone | Coverage (%) ↑ | Accuracy (Pass / Vote) (%) ↑ | #Files ↑ | nLOC ↑ | Code Tokens ↑ |
|---|---|---|---|---|---|---|
| Gold Projects (Reference) | Human Developers | 100.0 | 94.8 / 98.8 | 345 | 97,725 | 718,946 |
| ZeroRepo-Doc (Baseline) | GPT-4.1 | 64.6 | 50.0 / 63.4 | 209 | 6,079 | 158,948 |
|  | GPT-5-mini | 74.2 | 52.6 / 71.4 | 143 | 13,414 | 125,625 |
| ZeroRepo-RPG (Ours) | GPT-4.1 | **93.5** | **85.8 / 93.4** | **206** | **35,190** | **346,865** |
|  | GPT-5-mini | **98.5** | **86.0 / 97.7** | **226** | **60,871** | **550,432** |

*Table 3.* Ablation study of RPG-Encoder on SWE-bench Live. Best results (per backbone) are highlighted in **bold**.

| Backbone | Method | File-level | | | | Function-level | | | |
|---|---|---|---|---|---|---|---|---|---|
| | | Acc@1 | Acc@5 | Pre | Rec | Acc@1 | Acc@5 | Pre | Rec |
| GPT-4o | RPG-Encoder | **69.2** | **83.5** | **73.2** | **60.3** | **50.5** | **69.4** | **59.4** | **41.8** |
| | w/o Dependency | 58.4 | 77.4 | 63.0 | 53.3 | 44.8 | 66.3 | 53.4 | 36.4 |
| | w/o Feature | 60.9 | 76.3 | 64.6 | 52.4 | 43.1 | 63.4 | 52.3 | 35.5 |
| GPT-4.1 | RPG-Encoder | **78.0** | **90.5** | **81.4** | **69.0** | **64.7** | **81.9** | **72.1** | **52.6** |
| | w/o Dependency | 77.4 | 89.4 | 80.6 | 68.3 | 63.7 | 80.2 | 71.1 | 51.9 |
| | w/o Feature | 71.7 | 87.5 | 76.9 | 64.5 | 57.4 | 76.3 | 66.3 | 47.8 |

*Table 4.* Ablation on Representational Fidelity. We evaluate reconstruction performance on `scikit-learn` by progressively removing node-level metadata from the RPG using GPT-5-mini.

| Method | Coverage | Pass Rate | #Files | nLOC | Tokens |
|---|---|---|---|---|---|
| ZeroRepo-Docs | 72.3 | 55.6 / 66.3 | 76 | 12,007 | 101,988 |
| ZeroRepo-RPG (Ours) | **100.0** | **82.8 / 99.5** | **256** | **96,831** | **898,026** |
| w/o Function Metadata | 91.5 | 74.1 / 90.9 | 248 | 87,413 | 854,886 |
| w/o All Node Metadata | 87.2 | 65.3 / 84.7 | 157 | 63,489 | 687,879 |

ical boundaries, the model tends to merge distinct modules, leading to a loss of granularity. Additionally, the removal of function metadata (*w/o Function Metadata*) reduces the Pass Rate from 82.8% to 74.1%, showing that detailed structural signals are important for code correctness. Finally, all graph-based variants outperform the text-based ZeroRepo-Docs, confirming that structured representations provide a better basis for reconstruction than linear documentation.

# 7. Analysis

## 7.1. Representational Efficiency

*Table 5.* Efficiency for repository understanding on SWE-bench Verified. Steps and Cost are averaged over tasks. Eff. is defined as Acc@5/Cost. Additional results are provided in Appendix C.4.

| Method | GPT-4.1 | | | GPT-5 | | |
|---|---|---|---|---|---|---|
| | Steps | Cost ($) | Eff. | Steps | Cost ($) | Eff. |
| OrcaLoca | 20.22 | 0.46 | 1.48 | 38.60 | 0.82 | 1.06 |
| CoSIL | 19.77 | 0.24 | 3.10 | 19.52 | 0.31 | 2.64 |
| LocAgent | 11.94 | 0.86 | 0.76 | 6.48 | 0.49 | 1.64 |
| RPG-Encoder | **6.75** | **0.18** | **4.63** | **6.34** | **0.22** | **4.15** |

**RPG Facilitates Reasoning Efficiency.** Table 5 evaluates the efficiency of agents guided by different sub-

strates. Across all backbones, RPG-Encoder achieves fewer steps and lower expenditure, yielding the highest cost-effectiveness (Acc@5/Cost). On GPT-5, RPG-Encoder reaches an efficiency of 4.15 at a cost of $0.22, whereas baselines such as OrcaLoca and LocAgent require higher expenditures for lower efficiency gains. This trend is consistent with GPT-4.1 results, where RPG-Encoder attains the peak efficiency of 4.63. These results indicate that RPG-guided navigation enables precise exploration, concentrating reasoning resources on relevant code regions and reducing redundant API calls throughout the localization process.

## 7.2. Structural Evolvability

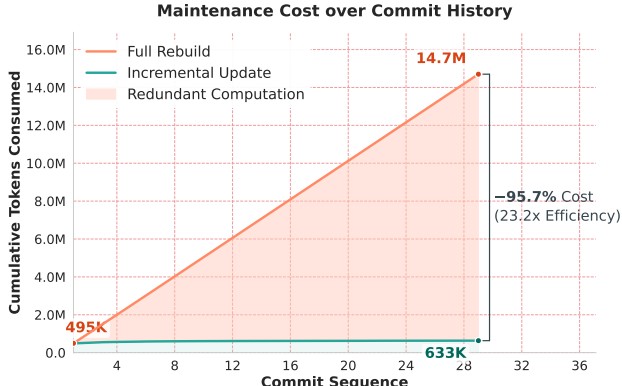

*Figure 3.* Cost Efficiency Comparison: Full RPG Rebuilding versus Incremental Diff-based Updates across Commit History.

**Incremental Maintenance Ensures Sustainable Scalability.** To assess feasibility, we measure maintenance costs across a commit sequence. Figure 3 shows that full reconstruction scales linearly and exceeds 14.7M tokens, whereas our incremental strategy uses only 633K tokens by isolating semantic deltas. This 95.7% reduction confines heavy computation to a one-time initialization and effectively decouples ongoing maintenance costs from repository scale, enabling sustainable long-term operation.

**Evolution Balance between Fidelity and Efficiency.** To validate resilience against semantic drift during updates, we assessed representational fidelity by deploying agents on SWE-bench Live using RPGs from both strategies. Table 6 indicates that the "Incr." strategy maintains statistical

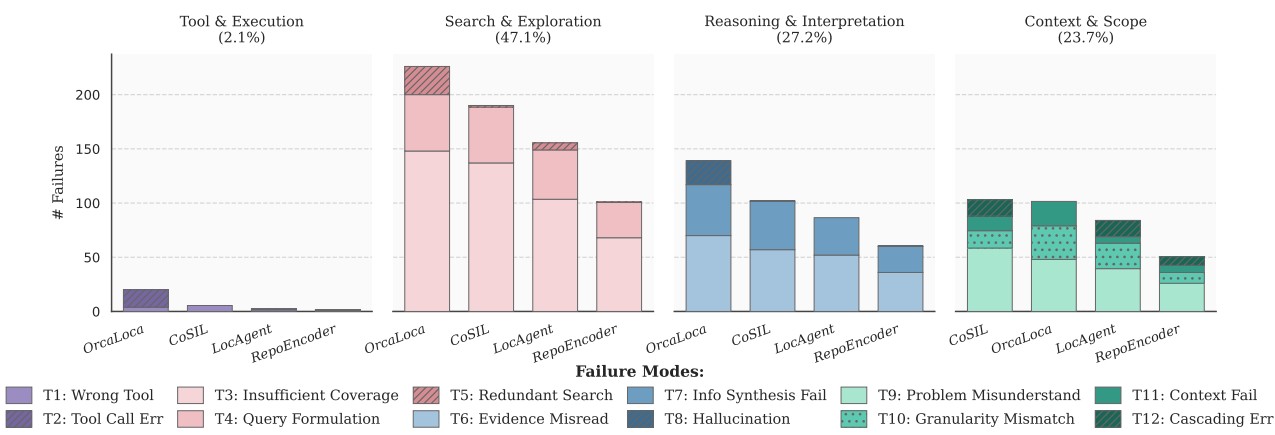

*Figure 4.* Distribution of Failure Modes on SWE-bench Verified. We analyze 100 sampled failed trajectories per method using GPT-4o. Errors are categorized into four macro-groups: Tool & Execution, Search & Exploration, Reasoning & Interpretation, and Context & Scope, with 12 fine-grained sub-types (T1–T12). More Details are in Appendix C.3

*Table 6.* Full vs. Incremental RPG Fidelity on SWE-bench Live. SWE-bench Live accuracy of RPGs across commits under full reconstruction (Full) and incremental maintenance (Incr.).

| Model | Strategy | File-level | | | | Function-level | | | |
|---|---|---|---|---|---|---|---|---|---|
| | | Acc@1 | Acc@5 | Pre | Rec | Acc@1 | Acc@5 | Pre | Rec |
| GPT-4o | Full | **69.9** | **84.6** | **73.2** | 60.1 | **53.8** | 68.5 | **60.6** | 41.1 |
| | Incr. | 69.2 | 83.5 | **73.2** | **60.3** | 50.5 | **69.4** | 59.4 | **41.8** |
| GPT-4.1 | Full | **79.9** | 88.2 | **82.5** | **69.8** | **67.4** | 80.3 | **73.3** | **55.4** |
| | Incr. | 78.0 | **90.5** | 81.4 | 69.0 | 64.7 | **81.9** | 72.1 | 52.6 |

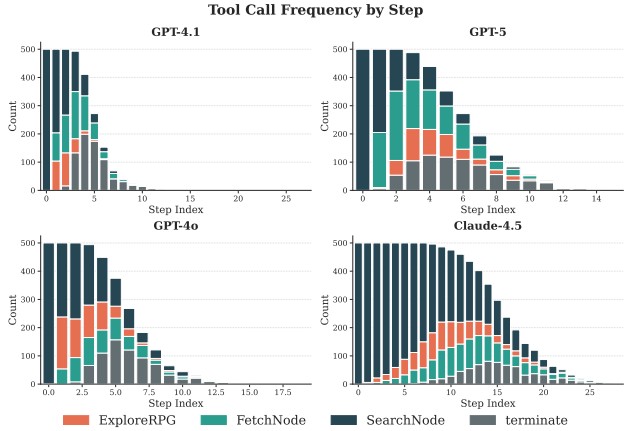

*Figure 5.* Impact of RPG Tooling on Agent Behavior on SWE-bench Verified. Step-wise action distributions induced by the RPG interface across LLMs.

parity with the "Full" baseline. Specifically, while "Incr." achieves slightly higher retrieval accuracy (81.9% Acc@5 compared to 80.3% for GPT-4.1), "Full" reconstruction retains a marginal edge, surpassing "Incr." by approximately 2% in Precision and Recall. This balance confirms that our sustainable evolution effectively preserves the repository's semantic integrity with negligible degradation.

### 7.3. Agentic Navigability

**RPG Induces Structured Exploration.** To investigate whether RPG structures reasoning, we visualized tool usage distributions across LLMs. Figure 5 reveals a universal "Search-then-Zoom" pattern: agents prioritize broad topology traversal (ExploreRPG, SearchNode) to establish a global map before narrowing to fine-grained analysis (FetchNode). This trend is more pronounced in stronger reasoners (e.g., Claude-4.5), which leverage RPG's structural context to support extended interaction horizons. These results confirm that RPG effectively guides agents from global comprehension to localized implementation.

**Dual-View Search Mitigates Navigational Failures.** We manually analyzed 100 failed trajectories from GPT-4o to identify error patterns mitigated by the RPG structure. As

shown in Figure 4, RPG reduces Search & Exploration failures compared to baselines. While systems like LocAgent and CoSIL utilize graph structures, they often suffer from Insufficient Coverage. RPG-Encoder addresses this by providing dual-path access, where semantic features enable broad global retrieval to expand the search space, while the structured hierarchy guides the agent to reduce Redundant Search. This multi-view navigation ensures agents can accurately localize intent before traversing implementation-level dependencies. Improved localization also reduces downstream errors in Context & Scope, keeping reasoning grounded in the correct implementation units.

## 8. Conclusion

In this work, we introduce RPG-Encoder, transforming the Repository Planning Graph (RPG) into a unified representation for repository reasoning. By coupling dense semantics

with topological constraints, RPG-Encoder bridges architectural intent and implementation. Our evaluations show RPG is a superior navigational map for localization and a blueprint for reconstruction, achieving significantly higher fidelity than documentation. Furthermore, our incremental mechanism ensures consistency with lower overhead. Ultimately, RPG-Encoder establishes a robust foundation for closed-loop software engineering by bidirectionally linking architectural intent with structural implementation.

## Impact Statement

This paper presents work whose goal is to advance the field of Machine Learning. There are many potential societal consequences of our work, none of which we feel must be specifically highlighted here.

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

# A. Detailed Methodology of RPG-Encoder

This section provides a deep dive into the implementation details and algorithmic foundations of RPG-Encoder, expanding upon the three core phases—Construction, Evolution, and Operation—introduced in Section 3 of the main text.

## A.1. RPG Extraction: Semantic Lifting and Hierarchical Encoding

This subsection details the construction stage of RPG-Encoder, which transforms a raw repository into a hierarchically organized *feature space* (the semantic backbone of RPG) together with a *grounded* mapping that links abstract functional nodes to concrete directory scopes. Concretely, the extraction stage proceeds in three steps: (1) **Semantic lifting** that converts low-level code entities into atomic functional features; (2) **Latent architecture recovery** that reorganizes these features into a consistent three-level hierarchy; and (3) **Artifact grounding** that anchors each abstract subtree to a compact set of physical directory paths. The resulting hierarchy serves as the *Functionality SubGraph* used by downstream agentic tools (Appendix A.3).

### A.1.1. SEMANTIC LIFTING VIA PROMPTED SEMANTIC PARSING

**Global parsing strategy.** Given a repository $\mathcal{R}$, semantic lifting is performed from a *global perspective* rather than on individual files in isolation. We first identify all code entities of interest, including classes, methods and functions, and treat them as the fundamental semantic units to be analyzed. This global view allows the model to maintain consistent semantic granularity across the repository and reduces local biases introduced by file boundaries.

**Semantic units and batching.** To accommodate repositories of varying scales while respecting model context limits, code entities are abstracted into *semantic units* and analyzed in batches under a controlled token budget. Each semantic unit represents a coherent functional entity, ensuring that semantically coupled components are interpreted in context. Batches are constructed to balance completeness and efficiency, such that every semantic unit is analyzed exactly once while enabling scalable processing of large repositories.

**Semantic feature representation.** For each code entity $u$, the parser produces a set of *atomic semantic features* $f(u) = \{a_1, a_2, \dots\}$, where each $a_i$ is a short verb–object phrase describing *what* the entity does rather than *how* it is implemented. These atomic features are intentionally constrained to be: (i) **single-responsibility**, (ii) **implementation-agnostic**, and (iii) **lexically normalized** (lowercase English, concise phrasing). This normalization is critical for subsequent routing and hierarchical encoding, since it provides stable semantic anchors for grouping and comparison across the repository.

**Prompt template (semantic parsing).** We implement semantic lifting using the following prompt template, which enforces: (1) complete coverage of all functions in the chunk, (2) strict output schema, and (3) feature naming rules that avoid vague verbs and implementation details. The prompt returns a JSON object mapping each function name to a list of semantic features.

---

**Semantic Parsing Prompt**

```
## Instruction
You are a senior software analyst.
Your goal is to analyze all functions in the current input and return their key semantic features  what each
function does, not how its implemented.

### Key Goals
- Complete analysis: Provide semantic feature extraction for every function in the given input. Do not skip any
function.
- Batch perspective: Analyze all functions in the chunk together, considering their roles within the overall
system.
- High-level behavior: Focus on the purpose and role of each function, not on low-level implementation details.
- If multiple definitions share the same method name (e.g., property getter and setter for the same attribute),
you may output that method name only once and merge their semantic features; you do not need to distinguish
decorator variants.

## Feature Extraction Principles
Follow these principles when analyzing functions:
1. Focus on the purpose and behavior of the function  what role it serves in the system.
2. DO NOT describe implementation details, variable names, or internal logic such as loops, conditionals, or
data structures.
3. If a function performs multiple responsibilities, break them down into separate features.
```

---

```
4. Use your understanding of each functions name, signature, and code to infer its intent.
5. Only analyze functions included in the current input  do not guess or invent other functions.
6. Do not omit any function, including utility or helper functions.

### Feature Naming Rules:
1. Use verb + object format (e.g., `load config`, `validate token`).
2. Use lowercase English only.
3. Describe purpose not implementation (focus on what, not how).
4. Each feature must express one single responsibility.
5. If a method has multiple responsibilities, split into multiple atomic features.
6. Keep features short and atomic (prefer 38 words; no full sentences; no punctuation).
7. Avoid vague verbs (`handle`, `process`, `deal with`); prefer precise verbs (`load`, `validate`, `convert`, `
update`, `serialize`, `compute`, `check`, `transform`).
8. Avoid implementation details (no loops, conditionals, data structures, control flow).
9. Avoid libraries/frameworks/formats (say `serialize data`, not `pickle object` / `save to json`).
10. Prefer domain/system semantics over low-level actions (`manage session` > `update dict`).
11. Avoid chaining actions (dont write `initialize config and register globally`; split into separate features).

## Output Format
You must respond with the following structure:
A `<solution>` block  a JSON object mapping each function name to a list of its semantic features
If a function does not implement any meaningful features (e.g., it's a stub), still include it with an empty
list.
### Output Template:
<solution>
{{
  "func_name_1": ["feature one", "feature two"],
  "func_name_2": [],
  ...
}}
</solution>

## Input Context
### Repository Name
<repo_name>
{repo_name}
</repo_name>
### Repository Overview
<repo_info>
{repo_info}
</repo_info>
```

**Post-processing and validation.** We apply lightweight validation to guarantee the output is machine-consumable: (i) JSON parsing and schema checking (every function in the input must appear as a key); (ii) feature list normalization (whitespace, casing, deduplication); and (iii) optional merging for decorator variants (e.g., property getter/setter) when they share the same method name, as allowed by the prompt. If the model returns malformed output, we retry with a minimal format correction instruction without changing semantic constraints.

**Illustrative example.** Figure 6 shows an end-to-end example of semantic lifting, where raw code snippets are mapped to their corresponding atomic semantic features.

A.1.2. LATENT ARCHITECTURE RECOVERY FOR HIERARCHICAL ENCODING

**Motivation.** Semantic lifting yields a set of fine-grained features distributed across many files, which is insufficient as a planning substrate: flat features are hard to navigate, while directory-only grouping often overlooks logical roles. We therefore recover a *latent functional architecture* that reorganizes the repository into a consistent, interpretable, and searchable hierarchy. We enforce a strict **three-level** feature path format:

$$<\text{functional area}>/<\text{category}>/<\text{subcategory}>,$$

which balances abstraction (top-level intent) and specificity (fine-grained specialization), while keeping routing and tool-based navigation tractable.

**Step 1: Domain discovery (functional areas).** We first discover a small set of high-level functional areas that act as architectural centroids. The model is guided to propose meaningful areas (e.g., `DataProcessing`, `ModelTraining`, `EvaluationMetrics`) while avoiding low-signal directories such as vendor code, tests, or documentation.

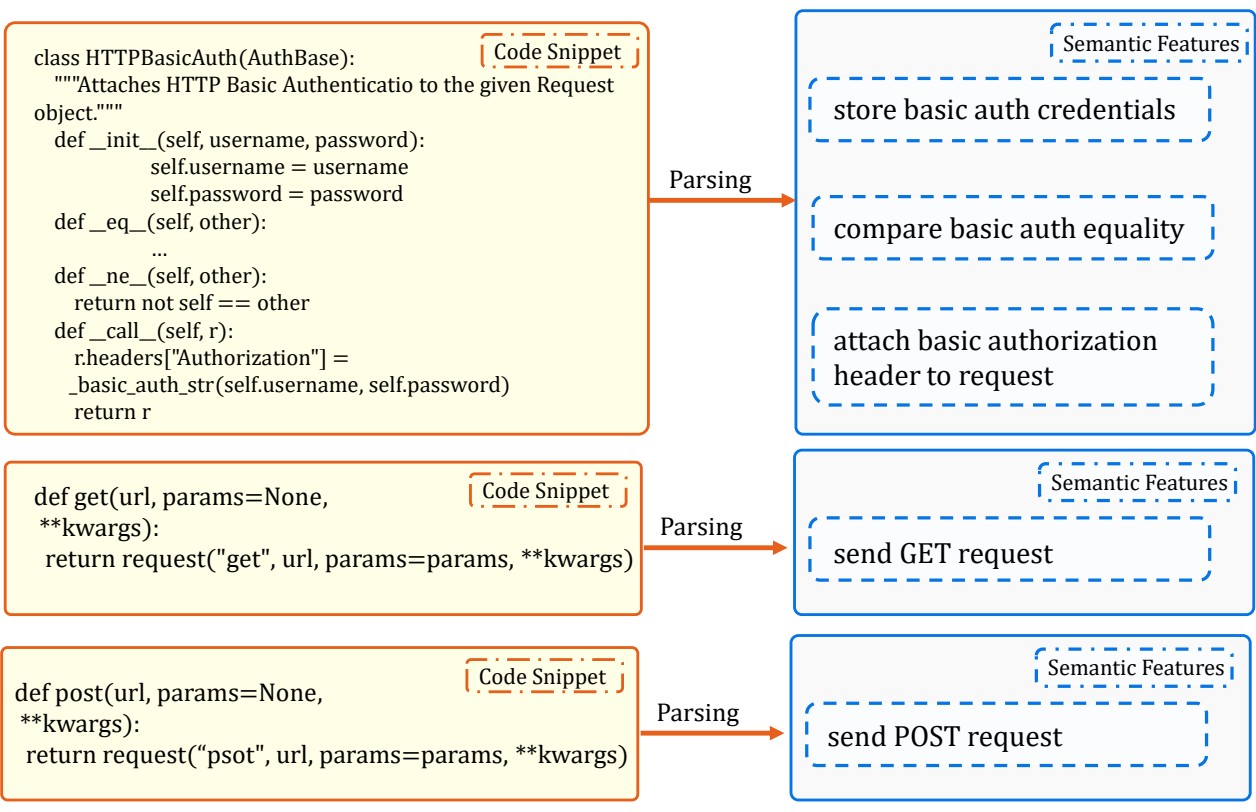

*Figure 6.* Illustration of raw code snippets and their corresponding semantic features extracted via semantic parsing.

Domain Discovery Prompt

```
## Instructions
You are an expert software architect and repository analyst.
You goal is to analyze the repository holistically and identify its main functional areas  coherent, high-level
modules or subsystems that reflect the repositorys architecture and purpose.

### Guidelines
- Think from a software architecture perspective; group code into major, distinct responsibilities (e.g., data
loading/processing, training/inference, evaluation/metrics, visualization/reporting, APIs/interfaces,
configuration/utilities/infrastructure).
- Avoid listing individual files or small helpers, third-party/vendor code, and build/test/docs directories.
- Ensure each area is meaningful and represents a clear responsibility in the codebase.

### Naming Principles
- Single Responsibility: Each area should cover one logical concern (e.g., "DataProcessing", "ModelTraining").
- High-Level Abstraction: Group related submodules; separate distinct layers.
- Consistency: Use PascalCase for names (e.g., "FeatureExtraction", "EvaluationMetrics").
- Meaningful Scope:
  - Merge closely related components (e.g., "data_loader", "dataset"  "DataProcessing")
  - Avoid vague terms like "core", "misc", "other"
  - Use domain-specific names when appropriate (e.g., "TextPreprocessing", "ImageSegmentation")

### Output Format
Return only the result in this exact format:
<solution>
[
"functional_area1", "functional_area2", "functional_area3", ...
]
</solution>
```

**Step 2: Hierarchical construction (three-level paths).**   Given the discovered functional areas and the parsed feature groups, we perform hierarchical construction by assigning each top-level feature group to a unique three-level target path. This step is formulated as a constrained semantic assignment problem: the model must use only the provided functional areas for the first level, and it must generate intent-focused category/subcategory labels following the same semantic naming rules used in semantic lifting.

Hierarchical Construction Prompt

```
## Instruction
You are an expert software architect and large-scale repository refactoring specialist.

## Goal
Reorganize and enrich the repositorys parsed feature tree by assigning each top-level feature group
(e.g., "data_loader", "model_trainer", "metrics") to the most semantically appropriate location
within the target architecture.

## Target Path Format (STRICT)
Each target path must have exactly three levels:
`<functional_area>/<category_level_1>/<subcategory_level_2>`
- `functional_area` must be one of the provided <functional_areas>.
- `category_level_1` expresses broader purpose or lifecycle role.
- `subcategory_level_2` adds precise specialization or context.
- Each segment: concise (25 words), semantically meaningful, intent-focused.
Examples:
- "data ingestion/pipeline orchestration/task scheduling"
- "model training/optimization strategy/hyperparameter tuning"
Avoid filler labels (e.g., "misc", "others", "core", "general").

## Semantic Naming Rules
When creating or adjusting semantic labels (categories/subcategories), follow:
1. Use "verb + object" phrasing; e.g., `load config`, `validate token`.
2. Use lowercase English only.
3. Describe purpose, not implementation.
4. Ensure each label expresses a single responsibility.
5. When multiple distinct roles exist, use multiple precise labels rather than one overloaded label.
6. Avoid vague verbs such as `handle`, `process`, and `deal with`.
7. Avoid implementation details, including control-flow or data-structure references.
8. Avoid mentioning specific libraries, frameworks, or formats; prefer `serialize data` over `pickle object` or
`save to json`.
9. Prefer domain or system semantics over low-level actions; use `manage session` rather than `update dict`.

## Scope Constraints
```

```
- Only assign top-level groups (keys of <parsed_folder_tree>).
- Exclude docs/examples/tests/vendor code unless essential to core functionality.
- Do not invent new functional areas; use only those in <functional_areas>.
- You may define new categories/subcategories as needed, but they must remain meaningful and consistent.

## Output Format (STRICT)
Return only the JSON object wrapped exactly as:
<solution>
{
  "<functional_area>/<category>/<subcategory>": ["top_level_group_1", "top_level_group_2", ...],
  "<functional_area>/<category>/<subcategory>": ["top_level_group_3", ...]
}
</solution>
```

**Outputs and usage.** The output of hierarchical construction is a mapping from feature paths to sets of feature groups, which induces a topological feature tree $\mathcal{T}_{\text{feature}}$. This tree serves two purposes: (i) it provides high-signal *search scopes* for intent-to-code mapping, and (ii) it supports routing and traversal by ensuring semantically coherent boundaries at each level.

**Illustrative examples.** Figures 7 and 8 provide examples of domain discovery and hierarchical construction, respectively.

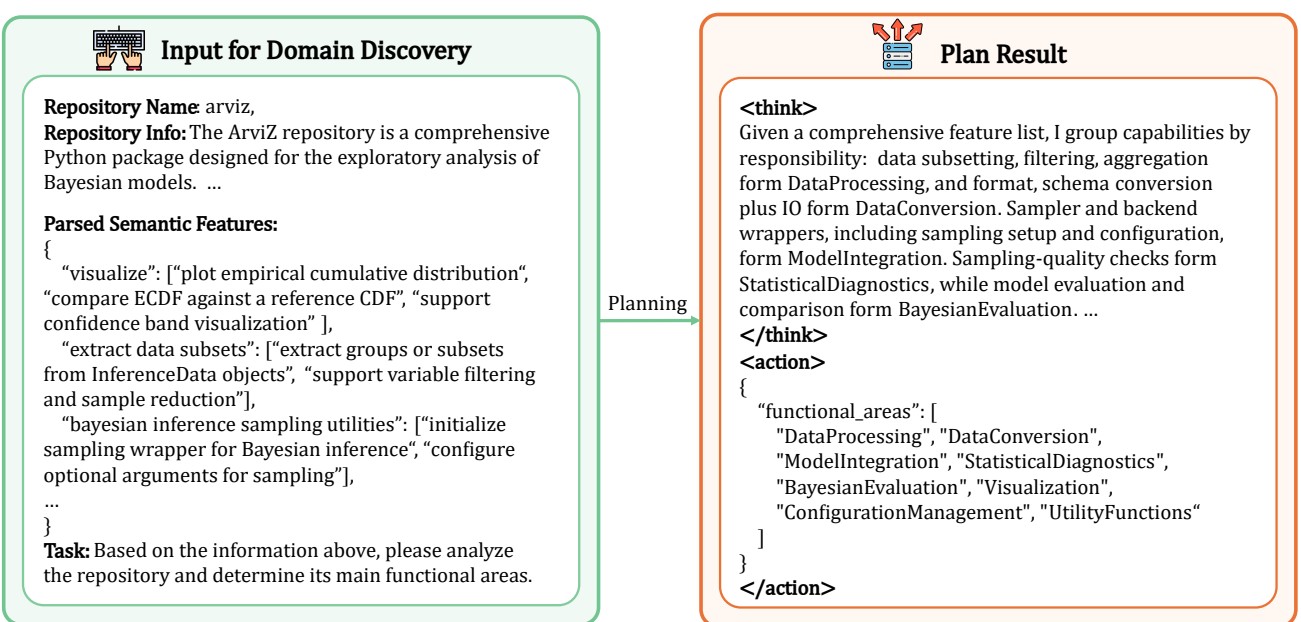

*Figure 7.* Illustrative example of the Domain Discovery phase.

### A.1.3. ARTIFACT GROUNDING: ANCHORING ABSTRACT SUBTREES TO DIRECTORY SCOPES

**Problem formulation.** To bridge the semantic hierarchy $\mathcal{T}_{\text{feature}}$ with physical repository artifacts, we ground each abstract node $v$ to a compact set of directory scopes. Let $\mathcal{L}(v)$ denote the set of leaf nodes in the subtree rooted at $v$. For each leaf node $l \in \mathcal{L}(v)$, let path$(l)$ be its physical file path. We define the **File Coverage** $\mathcal{C}(v)$ as the collection of parent directories for all leaves under $v$:

$$\mathcal{C}(v) = \{\text{dir}(\text{path}(l)) \mid l \in \mathcal{L}(v)\}, \tag{1}$$

where dir$(\cdot)$ extracts the directory component of a file path. We seek a compact representation $\hat{\pi}(v)$ that succinctly covers $\mathcal{C}(v)$ while preserving functional boundaries across distinct modules.

**Bottom-up propagation with Trie-based branching analysis.** A naive common-prefix (LCA) computation may over-collapse unrelated modules into overly general roots (e.g., /). To avoid this, we compute $\hat{\pi}(v)$ via a bottom-up propagation strategy that aggregates coverage and then simplifies it using a Trie-based branching analysis: all paths in $\mathcal{C}(v)$ are inserted into a Prefix Tree, and only **branching nodes** (multiple children or path termination) are retained as grounded scopes.

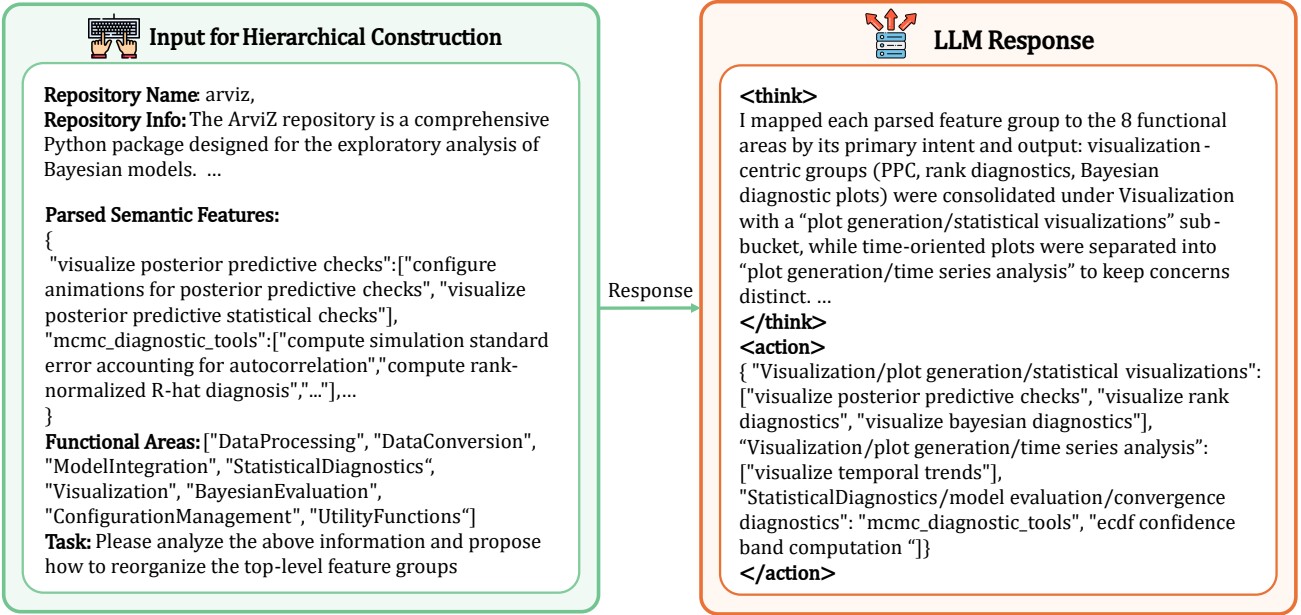

*Figure 8.* Example of the Hierarchical Construction phase.

This yields a minimal set of directory LCAs that dominate disjoint coverage regions while respecting module boundaries. Algorithm 1 provides the full procedure.

**Complexity analysis.** Let $N$ be the number of paths and $L$ the maximum directory depth. Trie construction and branching-node extraction take $O(N \cdot L)$ time, bounded by total path characters. Since propagation visits each feature node once in a bottom-up pass, the total grounding overhead scales linearly with repository size and is negligible compared with LLM inference.

## A.2. Incremental Evolution: Differential Update and Maintenance

This subsection details how RPG-Encoder maintains the Repository Planning Graph (RPG) under continuous codebase evolution. Given a repository update (e.g., a commit), our goal is to *incrementally* update the semantic hierarchy and its grounded mapping, ensuring the RPG remains a faithful semantic reflection of the codebase while avoiding expensive full reconstruction. We formulate repository evolution as a stream of **atomic operations**: DELETE, MODIFY, and INSERT. Each operation updates both (i) the local semantic representation of affected entities and (ii) their placement within the feature hierarchy.

### A.2.1. DIFFERENTIAL EVENT DETECTION AND OPERATION SCHEDULING

**From code diffs to semantic events.** Given a code change $\Delta$ (e.g., a git diff between two revisions), we extract changed code entities at the function/method granularity whenever possible. Each affected entity $u$ is categorized into one of three evolution events:

- **Deletion:** $u$ is removed from the repository.

- **Modification:** $u$ exists in both revisions but its implementation changes.

- **Insertion:** $u$ is newly introduced in the new revision.

For MODIFICATION, we further distinguish between semantically stable edits and substantial semantic drift (Section A.2.3), which determines whether the update can be handled locally or requires structural relocation.

---

**Algorithm 1** Bottom-Up Path Metadata Propagation

---

**Require:** Feature Tree $T = (V, E)$, Leaf paths $\text{path}(\cdot)$
**Ensure:** Grounded path assignments $\hat{\pi}(v)$ for all $v \in V$

 1: **Function** PROPAGATE($v$):
 2: **if** $v$ is Leaf **then**
 3:     **return** $\{\text{dir}(\text{path}(v))\}$             ▷ Base case: Return physical directory
 4: **end if**
 5: $\mathcal{S} \leftarrow \emptyset$
 6: **for** $child \in \text{Children}(v)$ **do**
 7:     $\mathcal{S} \leftarrow \mathcal{S} \cup \text{PROPAGATE}(child)$            ▷ Recursively aggregate child coverage
 8: **end for**
 9: $\hat{\pi}(v) \leftarrow \text{COMPUTELCA}(\mathcal{S})$          ▷ Abstract concrete paths into logical scopes
10: **return** $\mathcal{S}$          ▷ Propagate full coverage to upper layers
11: **End Function**

12: **Function** COMPUTELCA($\mathcal{S}$):
13: $Trie \leftarrow \text{BUILDTRIE}(\mathcal{S})$          ▷ Construct Prefix Tree from path set
14: $L \leftarrow \emptyset$
15: **for** $node \in \text{POSTORDER}(Trie)$ **do**          ▷ Bottom-up traversal for optimal pruning
16:     **if** $node.\text{is\_branching}()$ **or** $node.\text{is\_terminal}()$ **then**
17:         $L.\text{add}(node.\text{path})$          ▷ Identify meaningful functional boundary
18:         PRUNESUBTREE($node$)          ▷ Consolidate redundant sub-paths
19:     **end if**
20: **end for**
21: **return** $L$
22: **End Function**

---

**Scheduling principle.** We schedule evolution operations under constraints that preserve structural consistency of the hierarchy, prevent intermediate abstract nodes from accumulating dead branches, and ensure that newly introduced entities do not disrupt the existing topological organization.

A.2.2. NODE DELETION WITH STRUCTURAL HYGIENE

**Motivation.** Deletion must maintain structural integrity of the hierarchy. Removing a leaf entity may render its ancestor abstract nodes semantically vacuous (i.e., nodes that no longer cover any concrete code entities). Without cleanup, these dead branches accumulate and reduce the signal-to-noise ratio for search and routing.

**Recursive pruning.** We enforce **structural hygiene** via bottom-up pruning: after removing a leaf node, we recursively delete any ancestor abstract node whose subtree becomes empty. Pruning terminates once an ancestor still has remaining children or once the root is reached. This mechanism prevents stale semantic categories from persisting after refactors and ensures that the hierarchy remains compact and representative of the current repository state.

**Algorithm.** Algorithm 2 specifies the deletion procedure and the recursive pruning logic.

A.2.3. DIFFERENTIAL MODIFICATION PROCESSING

**Motivation.** A code edit may either preserve the original intent (e.g., bug fixes, refactoring, parameter tuning) or substantially change functionality (semantic drift). Treating both cases identically is suboptimal: in-place updates are sufficient for minor edits, while major drift requires relocating the entity to a semantically congruent domain.

**Minor update vs. semantic drift.** Given a modified entity $u$ with old/new versions $(u^{old}, u^{new})$, we compute semantic features $f(u^{old})$ and $f(u^{new})$ using the same parsing constraints as in extraction. We then assess drift based on: (i) feature overlap/consistency, and (ii) an LLM judgement constrained by explicit criteria. If drift is minor, we perform an in-place update of the node's semantic summary; otherwise, we trigger re-routing.

---

**Algorithm 2** Incremental Deletion (Recursive Pruning)

---

**Require:** Current graph $G$, target node id $id$
**Ensure:** Updated graph $G$
 1: **function** DELETENODE($G, id$)
 2:     $v \leftarrow$ GETNODE($G, id$)
 3:     **if** $v = \bot$ **then**
 4:         **return** $G$
 5:     **end if**
 6:     $parent \leftarrow v.$parent
 7:     REMOVENODE(G, v)                          ▷ Remove node and incident edges
 8:     PRUNEORPHANS(G, parent)                  ▷ Structural hygiene
 9:     **return** $G$
10: **end function**
11: **function** PRUNEORPHANS($G, v$)
12:     **if** $v = \bot \ \vee$ ISROOT($v$) **then**
13:         **return**
14:     **end if**
15:     **if** ISEMPTY($v.$children) **then**
16:         $gp \leftarrow v.$parent
17:         REMOVENODE(G, v)                 ▷ Prune empty abstract category
18:         PRUNEORPHANS(G, gp)                ▷ Recurse upwards
19:     **end if**
20: **end function**

---

**Re-routing as composition.** When semantic drift is significant, we treat modification as a composition of atomic operations:

$$\text{MODIFY}(u) \ \Rightarrow \ \text{DELETE}(u^{old}) + \text{INSERT}(u^{new}),$$

which relocates the entity to a new functional domain via the same semantic routing procedure used for insertion. This guarantees that the hierarchy reflects the updated intent rather than only updating text summaries in an incorrect domain.

**Algorithm.** Algorithm 3 formalizes the differential modification procedure, including the branching logic between in-place update and re-routing.

### A.2.4. NODE INSERTION VIA SEMANTIC ROUTING

**Motivation.** Naively attaching new entities to the root (or a fixed default module) breaks the semantic organization of the RPG and degrades downstream navigation. Instead, we treat insertion as a **semantic placement** problem: find the most appropriate abstract parent node in the current feature hierarchy that best matches the new entity's functionality.

**Routing objective.** Let $u$ be a newly added code entity with semantic features $f(u)$. Starting from the root of the feature hierarchy, we iteratively select the child domain whose description best aligns with $f(u)$, drilling down until no more meaningful specialization is possible. This **top-down semantic routing** ensures that $u$ is inserted into the most specific functional domain available while preserving interpretability of the hierarchy.

**Algorithm.** Algorithm 4 formalizes the insertion procedure. At each step, the router considers the candidate children of the current node and chooses the best target; if no child is sufficiently compatible, the algorithm terminates and inserts $u$ at the current level. This prevents over-forcing entities into unrelated subtrees.

**Complexity and scalability.** Incremental evolution in RPG-Encoder is inherently local. Each atomic operation affects only a bounded region of the hierarchy, without requiring global reconstruction. As a result, maintenance cost scales with the *magnitude of the change* rather than the size of the repository, enabling efficient and stable synchronization of the RPG under continuous development.

---

**Algorithm 3** Differential Modification Handling

---

**Require:** Graph $G$, file $f$, diff $\Delta$
**Ensure:** Updated graph $G$
 1: **function** PROCESSMODIFICATION($G, f, \Delta$)
 2:     $\langle \mathcal{U}^+, \mathcal{U}^-, \mathcal{U}^\sim \rangle \leftarrow$ PARSEUNITDIFF($\Delta$)
     **1) Delete / Insert**
 3:     **for all** $u \in \mathcal{U}^-$ **do**
 4:         $G \leftarrow$ DELETENODE($G, u.\text{id}$)
 5:     **end for**
 6:     **for all** $u \in \mathcal{U}^+$ **do**
 7:         $G \leftarrow$ INSERTNODE($G, u,$ LLMEXTRACT($u$))
 8:     **end for**
     **2) Update / Re-route**
 9:     **for all** $u \in \mathcal{U}^\sim$ **do**
10:         $v \leftarrow$ GETNODE($G, u.\text{id}$); $v.f \leftarrow$ LLMUPDATE($u$)
11:         **if** SEMANTICSHIFT($v$) $> \tau_{\text{drift}}$ **then**
12:             $G \leftarrow$ DELETENODE($G, u.\text{id}$)                    ▷ logic drift
13:             $G \leftarrow$ INSERTNODE($G, u,$ LLMEXTRACT($u$))
14:         **end if**
15:     **end for**
16:     **return** $G$
17: **end function**

---

## A.3. RPG Operation: Agentic Tool-use and Navigation Logic

This subsection details how RPG is operationalized as an actionable substrate for repository understanding. Beyond serving as a semantic representation, RPG exposes a *tool interface* that bridges high-level intents to concrete code entities and their dependency contexts. Concretely, we provide three complementary tools: **SearchNode** for intent-based discovery, **FetchNode** for precision context retrieval, and **ExploreRPG** for structural traversal on the RPG topology.

### A.3.1. TOOL INTERFACES AND PROMPT SPECIFICATIONS

**Design principles.** The tool suite is designed to support a common agent workflow in repository understanding: (i) start from vague or behavioral intents and obtain candidate code anchors; (ii) verify anchors with precise source context; and (iii) expand locally to cover call chains and related components. To ensure tool outputs are deterministic and machine-consumable, each tool prompt defines a strict parameter schema and return format.

**SearchNode: intent-based discovery.** **SearchNode** unifies *semantic discovery* and *textual retrieval*. It supports three modes: `features` (intent → feature nodes / mapped code entities), `snippets` (keyword/symbol search over the repository), and `auto` (feature mapping first, followed by snippet search when needed). Importantly, `search_scopes` can restrict the search to selected feature subtrees, leveraging the grounded hierarchy constructed in Appendix A.1 to improve precision.

---

**SearchNode Tool Prompt**

```
## Tool Name: SearchNode
### Description
Unified search tool for repository navigation. Use it to (1) map high-level functional/behavioral descriptions
to concrete code entities via RPG mapping, and/or (2) retrieve concrete code snippets via symbol/file/keyword
search. Prefer behavior-to-code mapping when you don't know the exact file/class/function name; then narrow down
 with snippet search.
Tip: Avoid vague terms; use concrete behavior phrases or high-signal identifiers.
### Parameters
{
  "tool_name": "SearchNode",
  "parameters": {
```

---

---

**Algorithm 4** Incremental Additions (LLM-Based Semantic Routing)

---

**Require:** Current Graph $G$, New Unit $u$, Feature Summary $f_u$
**Ensure:** Updated Graph $G$ with $u$ inserted
  1: **function** INSERTNODE($G, u, f_u$)
  2:      $v_{\text{best}} \leftarrow$ FINDBESTPARENT($G.\text{root}, f_u$)
  3:      $v_{\text{new}} \leftarrow$ CREATENODE($u, f_u$)
  4:      ADDEDGE($G, v_{\text{best}}, v_{\text{new}}, \mathcal{E}_{\text{feature}}$)                           ▷ Attach to semantically determined parent
  5:      **return** $G$
  6: **end function**

  7: **function** FINDBESTPARENT($v_{\text{curr}}, f_{\text{target}}$)
  8:      $Candidates \leftarrow \{ c \in$ CHILDREN($v_{\text{curr}}$) $\mid$ ISABSTRACT($c$) $\}$
  9:      **if** $Candidates = \emptyset$ **then**
 10:          **return** $v_{\text{curr}}$                            ▷ Base case: No deeper abstract categories
 11:      **end if**
 12:                                          ▷ Prompt LLM to select the best functional fit among children
 13:      $Context \leftarrow \{(c.\text{id}, c.f) \mid c \in Candidates\}$
 14:      $v_{\text{choice}} \leftarrow$ LLM_ROUTE($Context, f_{\text{target}}$)
 15:      **if** $v_{\text{choice}} \neq$ **null then**
 16:          **return** FINDBESTPARENT($v_{\text{choice}}, f_{\text{target}}$)         ▷ LLM chose a branch, drill down recursively
 17:      **else**
 18:          **return** $v_{\text{curr}}$                           ▷ LLM decided no child is a better fit
 19:      **end if**
 20: **end function**

```
    "mode": "<'features' | 'snippets' | 'auto'. Required. 'auto' may run both: feature-mapping first, then
snippet search.>",
    "feature_terms": "<List of concrete behavioral/functionality phrases. Required when mode is 'features' or '
auto'.>",
    "search_scopes": "<List of valid feature entity paths to restrict the Functionality SubGraph. Optional.>",
    "search_terms": "<List of file paths, qualified entities (file:Class.method), or high-signal text keywords.
Required when mode is 'snippets' or when 'auto' proceeds to snippet search.>",
    "line_nums": "<Two integers [start, end] to extract lines from a specific file. Requires an exact file path.
 Optional.>",
    "file_path_or_pattern": "<File path or glob pattern to restrict snippet search. Default: '**/*.py'>",
  }
}
### Returns
- If feature search runs: matched feature nodes mapped to code entities (feature name, code entity, file path,
line range when available)
- If snippet search runs: matched code snippets, complete files, or located entities based on search terms /
line ranges
```

**FetchNode: precision retrieval and verification.** **FetchNode** retrieves exact source context and metadata for known candidates (code entities or feature paths). It is used as a verification step after discovery to ensure the agent reasons on faithful code snippets rather than speculative guesses. FetchNode returns file paths, line ranges, entity types, mapped feature information, and a code preview.

FetchNode Tool Prompt

```
## Tool Name: FetchNode
### Description
- Retrieve precise metadata and source context for known code or feature entities.
- Use this tool to verify candidate code locations after identifying them through searches or graph exploration.
- Returns exact file path, entity type, start/end lines, mapped feature information, and a code preview.

### Parameters
{
  "tool_name": "FetchNode",
  "parameters": {
```

```
    "code_entities": "<List of existing and validated code entities in the current repository; non-existent
paths or speculative entities may be ignored. Optional.>",
    "feature_entities": "<List of existing and validated feature paths in the current repository; non-existent
entries may be ignored. Optional.>"
  }
}
### Returns
- Entity type (file/class/method/feature)
- Feature paths and code content (with source context / preview)
- Start/end lines and mapped feature information (when available)
```

**ExploreRPG: topological traversal.** ExploreRPG exposes the structural connectivity of RPG, enabling traversal along dependency edges (`imports`, `invokes`, `inherits`, etc.) and/or containment/composition relations. Starting from validated anchors, the agent can traverse upstream/downstream to uncover dependencies, impacted components, and semantically related regions.

### ExploreRPG Tool Prompt

```
## Tool Name: ExploreRPG
### Description
- Explore call chains and functional paths in the Repository Planning Graph.
- Starting from known code or feature entities, traverse upstream/downstream to discover related functions,
files, and feature nodes.
### Parameters
{
  "tool_name": "ExploreRPG",
  "parameters": {
    "start_code_entities": "<Optional list of existing code entities in the current repository (file paths,
classes, functions, or qualified names). Non-existent/speculative entities may be ignored or rejected.>",
    "start_feature_entities": "<Optional list of existing feature paths in the current repository. Non-existent
entries may be ignored or rejected.>",
    "direction": "<Traversal direction: 'upstream' (dependencies), 'downstream' (dependents), or 'both'. Default
: 'downstream'.>",
    "traversal_depth": "<Maximum traversal depth. Default: 2. Use -1 for unlimited depth.>",
    "entity_type_filter": "<Optional filter restricting traversal node types. Valid values: 'directory', 'file',
 'class', 'function', 'method'.>",
    "dependency_type_filter": "<Optional filter restricting dependency edge types. Valid values: 'composes', '
contains', 'inherits', 'invokes', 'imports'.>"
  }
}
### Returns
- Connected nodes and edges (code or feature view)
- Hints for invalid or fuzzy matches
```

A.3.2. TOOL-USE POLICY FOR REPOSITORY UNDERSTANDING

**Canonical tool orchestration.** We adopt a simple and robust orchestration policy that prioritizes semantic grounding before reading large contexts. Given a natural-language intent $\mathcal{I}$, the agent executes:

1. **Semantic discovery (SearchNode/features or auto):** convert $\mathcal{I}$ into concrete behavioral terms and retrieve candidate feature nodes and mapped code entities. If available, supply `search_scopes` to restrict discovery to the most relevant functional subtrees.

2. **Precision verification (FetchNode):** for top candidates, fetch exact code context (file path + line range + preview) and confirm semantic compatibility. Candidates that cannot be verified are discarded.

3. **Local expansion (ExploreRPG):** from verified anchors, traverse dependency edges (e.g., `invokes`, `imports`) to recover call chains, utilities, and related modules. This step is used to (i) locate the root cause, (ii) map the impact surface, or (iii) identify integration points.

4. **Pinpoint retrieval (optional SearchNode/snippets):** if the target remains ambiguous, run snippet search with high-signal identifiers obtained from previous steps (exact symbols, file paths, error strings), optionally extracting specific line ranges.

**Fallback rules.** When semantic discovery returns insufficient recall (e.g., missing/weak feature matches), the agent falls back to `snippets` mode to bootstrap concrete anchors, then returns to **FetchNode** and **ExploreRPG**. When snippet search yields too many matches, the agent tightens constraints by adding (i) feature scopes, (ii) file path patterns, or (iii) symbol-qualified queries.

This policy minimizes wasted context and reduces hallucination risk: SearchNode provides intent-to-code grounding, FetchNode ensures the agent reasons on exact source, and ExploreRPG reveals topological structure that cannot be reliably inferred from local snippets alone.

### A.3.3. EXECUTION TRACES AND EXAMPLES

We illustrate the practical efficacy of these tools through the execution traces shown in Figure 9. These traces demonstrate how the agent navigates from abstract intents to specific code implementations, leveraging both the semantic hierarchy and the dependency topology of RPG.

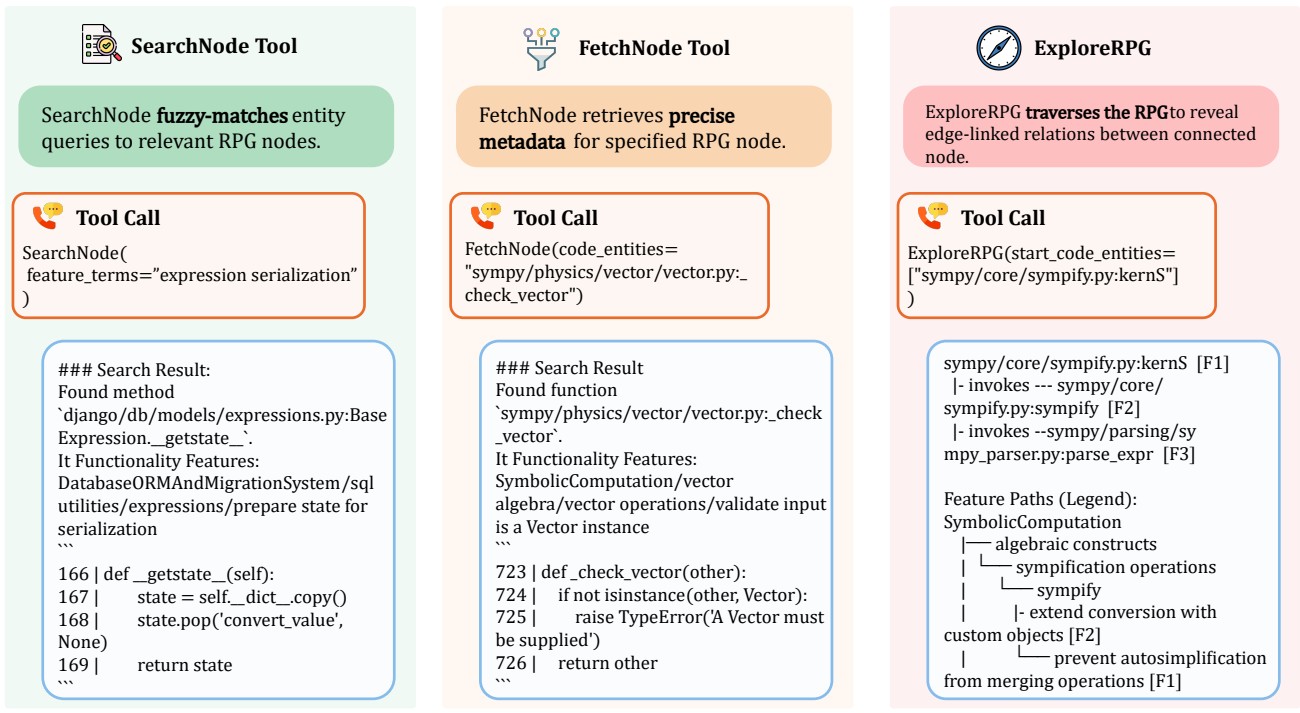

Figure 9. Execution traces of the three primary agentic tools. **SearchNode** maps abstract intent to concrete code; **FetchNode** retrieves precise source context; and **ExploreRPG** reveals topological connections and call relations.

As depicted in the figure, each tool provides distinct structural signals that support the agent's reasoning:

- **SearchNode (Left):** demonstrates intent-to-code grounding by mapping a behavioral query (e.g., "expression serialization") to a concrete code entity and its associated feature description. This step transforms ambiguous intent into executable anchors.

- **FetchNode (Center):** retrieves precise source context for a candidate entity (e.g., `_check_vector`), including exact line ranges and a preview snippet, enabling verification and preventing reasoning on speculative locations.

- **ExploreRPG (Right):** traverses the RPG topology from a verified anchor (e.g., `kernS`) to expose invocation and dependency relations. By showing edges such as `invokes` and their connected nodes, the agent can recover call chains and impacted modules, supporting systematic debugging and repository-level understanding.

# B. Experiment Setup

This appendix provides additional experimental setup details. It is organized into two parts: (i) repository reconstruction and (ii) repository understanding, including detailed baseline configurations and formal metric definitions.

## B.1. Repository Understanding

### B.1.1. EXPERIMENT SETUP

We describe the implementation details and baseline configurations for the repository understanding task. Our goal is to facilitate reproducibility and ensure fair and controlled comparisons across different localization pipelines.

**Common evaluation protocol.** All methods are evaluated under a shared protocol, including identical datasets, evaluation metrics, and termination criteria. Unless otherwise specified, we use the same preprocessing, canonicalization, and ranking-based evaluation procedures described in Section B.1.3.

**Backbone models.** We evaluate multiple large language model backbones to assess the robustness of each localization pipeline to the underlying model choice, including *o3-mini*(o3-mini-20250131) (OpenAI, 2025d), *GPT-4o*(gpt-4o-20241120) (OpenAI, 2024), *GPT-4.1*(gpt-4.1-20250414) (OpenAI, 2025a), *GPT-5*(gpt-5-20250807) (OpenAI, 2025b), *DeepSeek-V3.1* (Liu et al., 2024), and *Claude-Sonnet-4.5* (Anthropic, 2025b).

**Baselines.** We compare against representative repository-level localization pipelines: **Agentless** (Xia et al., 2024), **LocAgent** (Chen et al., 2025c), **CoSIL** (Jiang et al., 2025), and **OrcaLoca** (Yu et al., 2025). For each baseline, we retain the original algorithmic structure and design choices, making only minimal and necessary adaptations to the benchmark interface and backbone model to ensure compatibility with the shared evaluation protocol.

**Agentless** Agentless (Xia et al., 2024) employs a staged non-agentic workflow: (1) **Direct Prediction**: LLM predicts suspicious files directly from the issue description. (2) **Filtered Retrieval**: It performs embedding search within a search space pruned of "irrelevant folders." (3) **Candidate Aggregation**: Results from both streams are merged to maximize file-level recall.(4) **Element Localization**: Granularity is narrowed from files to specific code elements. (5) **Edit Localization**: The system pinpoints line-level edit targets within those elements. To ensure reproducibility, we apply specific parameter constraints corresponding to these stages. **Globally**, across all ranking steps, we maintain top_n=10 and enforce determinism via num_samples=1. **Stage-specific configurations** are set as follows: For **Retrieval (Step 2)**, we employ jinaai/jina-embeddings-v3 as the embedding backbone and set filter_type="given_files" to strictly enforce the LLM-generated folder constraints. For **Fine-grained Localization (Steps 4-5)**, we enable the --compress flag, which optimizes context utilization by condensing verbose code details while preserving salient information for precise element and edit identification.

**LocAgent** LocAgent (Chen et al., 2025c) is a dependency-graph integrated agent framework for repository-level localization, which wraps the dependency graph into three tools: (1) **SearchEntity**: searches relevant files/classes/functions from text queries (supports fuzzy match). (2) **TraverseGraph**: multi-hop traverses dependency relations from a seed entity to surface connected candidates. (3) **RetrieveEntity**: fetches the full metadata and code of selected entities for final inspection and ranking. For LocAgent, we do not impose any restriction on the number of iterative search rounds. To maximize the chance of producing a valid final prediction, we set the maximum retry budget to 3 attempts, and take the first well-formed output that satisfies the evaluation interface. We run LocAgent in **function-calling** mode with parallelism set to 1. We impose no explicit limit on the number of iterative search steps, and set the maximum retry budget to 3 to maximize the chance of producing a valid final output.

**CoSIL** CoSIL (Jiang et al., 2025) adopts a hybrid *agentic and workflow* strategy that explores code dependencies via iterative call-graph searching: it first performs broad exploration with a module call graph, then expands to a function call graph for deeper search, while using pruning and reflection to control direction and stabilize tool-formatted outputs. Following CoSIL's implementation details, We run CoSIL in **function-calling** mode with parallelism set to 1. We do not explicitly cap its iterative graph-search rounds, and allow up to 3 retries to maximize the chance of obtaining a valid final output.

**OrcaLoca** OrcaLoca (Yu et al., 2025) combines agentic code-graph exploration with a **dynamic-analysis** signal, using *bug reproduction* and *regression tests* to guide iterative search and candidate verification. It introduces two key mechanisms: (1) **Action decomposition** factorizes the large search action space into a hierarchical decision process (e.g., first selecting candidate classes, then narrowing to files), and applies top-$k$ selection for class decomposition and file decomposition; (2) **Distance-aware context pruning** retains only a fixed budget of the most relevant context entries (12 in our setup), prioritizing code units that are closer to the current targets in the dependency/call graph to improve context efficiency. We follow the original setup: for action decomposition, it applies top-$k$ selection with $k=3$ for class decomposition and $k=2$ for file decomposition, and uses distance-aware context pruning with a budget of 12 retained entries.

### B.1.2. EVALUATION TARGETS AT MULTIPLE GRANULARITIES

To assess localization quality at different levels of abstraction, we evaluate predictions at two granularities using a unified canonicalization scheme.

Each predicted or ground-truth location is mapped to a canonical string key through a granularity-specific formatter, ensuring consistent comparison across methods.

**File-level.** At the file level, a location is represented by its relative file path, *e.g.*, `path/to/file.py`. For an instance $i$, both the ground-truth set $G_i^{\text{file}}$ and the ranked prediction list $\pi_i^{\text{file}}$ consist of file paths. File-level evaluation measures whether a method can correctly identify the source files that contain the relevant implementation.

**Function-level.** At the function level, a location is represented by a fully qualified entity identifier within a file, formatted as `file:entity`. To avoid artificial mismatches caused by syntactic variations, constructor annotations are normalized by removing the suffix `.__init__` when present. For example, `a/b.py:Foo.__init__` is canonicalized to `a/b.py:Foo`.

Function-level evaluation assesses whether a method can precisely localize the relevant function or class definition beyond the file boundary.

When function-level annotations are unavailable for a given instance, we restrict the evaluation to the file level.

For both ground-truth and predicted locations, we remove duplicate entries while preserving their original order before computing all ranking metrics.

### B.1.3. METRICS

We formalize the ranking-based evaluation protocol for file-level localization. Let $\mathcal{I}$ denote the set of evaluation instances. For each instance $i \in \mathcal{I}$:

- $G_i$ denotes the set of ground-truth relevant files (or locations), with cardinality $|G_i| = m_i$.

- $\pi_i = (p_{i,1}, p_{i,2}, \ldots, p_{i,|\pi_i|})$ denotes the ranked prediction list produced by a method.

We define a binary hit indicator sequence $\mathbf{h}_i \in \{0, 1\}^{|\pi_i|}$ as

$$h_{i,j} = \begin{cases} 1, & \text{if } p_{i,j} \in G_i, \\ 0, & \text{otherwise,} \end{cases} \quad j = 1, \ldots, |\pi_i|. \tag{2}$$

All metrics are computed per instance and then averaged over $\mathcal{I}$.

**Accuracy@k (Acc@k).** Accuracy@k measures whether at least one ground-truth item appears within the top-$k$ predictions:

$$\text{Acc@k} = \frac{1}{|\mathcal{I}|} \sum_{i \in \mathcal{I}} \mathbb{I}\left[\sum_{j=1}^{k} h_{i,j} \geq 1\right]. \tag{3}$$

In our experiments, we report results for $k \in \{1, 3, 5\}$.

**Recall.** Recall measures the fraction of ground-truth items that are successfully retrieved by the model across the entire ranked list:

$$\text{Recall} = \frac{1}{|\mathcal{I}|} \sum_{i \in \mathcal{I}} \begin{cases} \frac{\sum_{j=1}^{|\pi_i|} h_{i,j}}{|G_i|}, & |G_i| > 0, \\ 0, & |G_i| = 0. \end{cases} \tag{4}$$

**Precision.** Precision measures the proportion of correct predictions among all retrieved items:

$$\text{Precision} = \frac{1}{|\mathcal{I}|} \sum_{i \in \mathcal{I}} \frac{\sum_{j=1}^{|\pi_i|} h_{i,j}}{|\pi_i|}. \tag{5}$$

### B.2. Details about Repository Reconstruction

In this section, we provide a comprehensive description of the experimental setup, workflow logic, and termination protocols for the two comparative settings: ZeroRepo-Doc and ZeroRepo-RPG.

#### B.2.1. REPOCRAFT BENCHMARK CONSTRUCTION

To rigorous evaluate the capabilities of automated repository reconstruction, we adapted the **RepoCraft** benchmark. The benchmark consists of real-world Python repositories selected for their popularity and structural complexity.

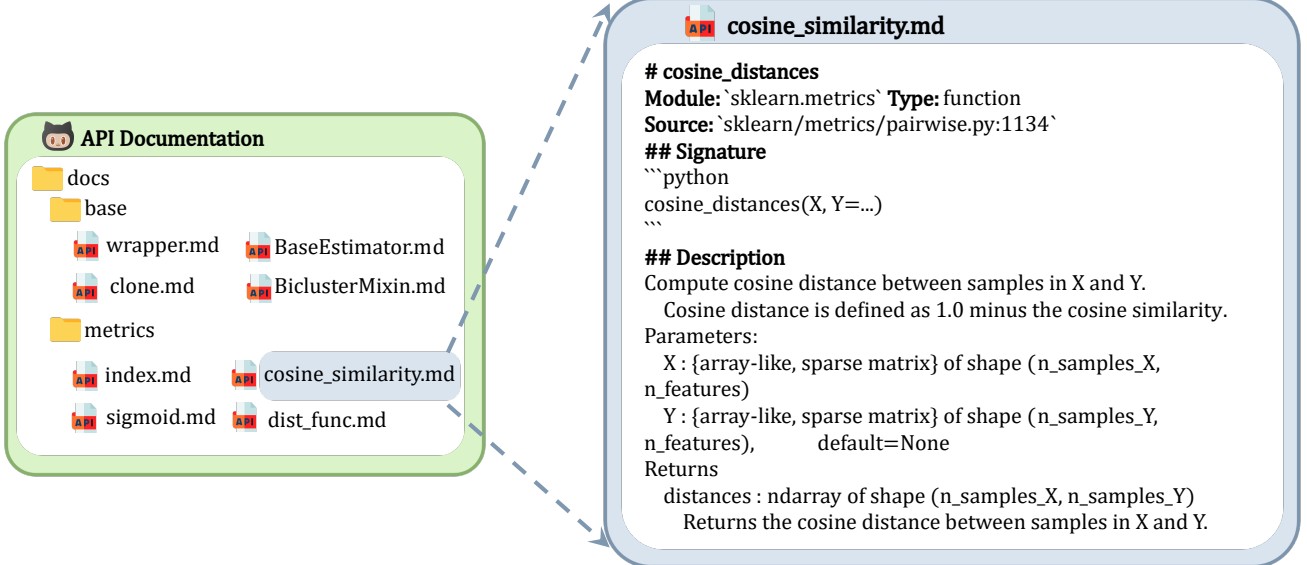

*Figure 10.* Illustration of the hierarchical organization of the API documentation (left) and an example of the granular content within a specific documentation node (right), detailing function signatures and parameter descriptions.

*Table 7.* Statistics of the compiled API documentation across the six subject repositories.

| Metric | Django | Pandas | Requests | Sklearn | Stats | SymPy |
|---|---|---|---|---|---|---|
| **# Files** | 2,863 | 1,536 | 15 | 1,052 | 1,244 | 610 |
| **# Tokens** | 374,586 | 1,314,314 | 52,351 | 235,319 | 405,022 | 180,505 |

**Documentation Compilation.** A critical component of our control setting is the provision of high-quality, official API documentation to serve as the ground-truth specification. We constructed this documentation dataset by processing the source files located in the `docs/` directory of each target repository. Specifically, we utilized **Sphinx**, the standard Python

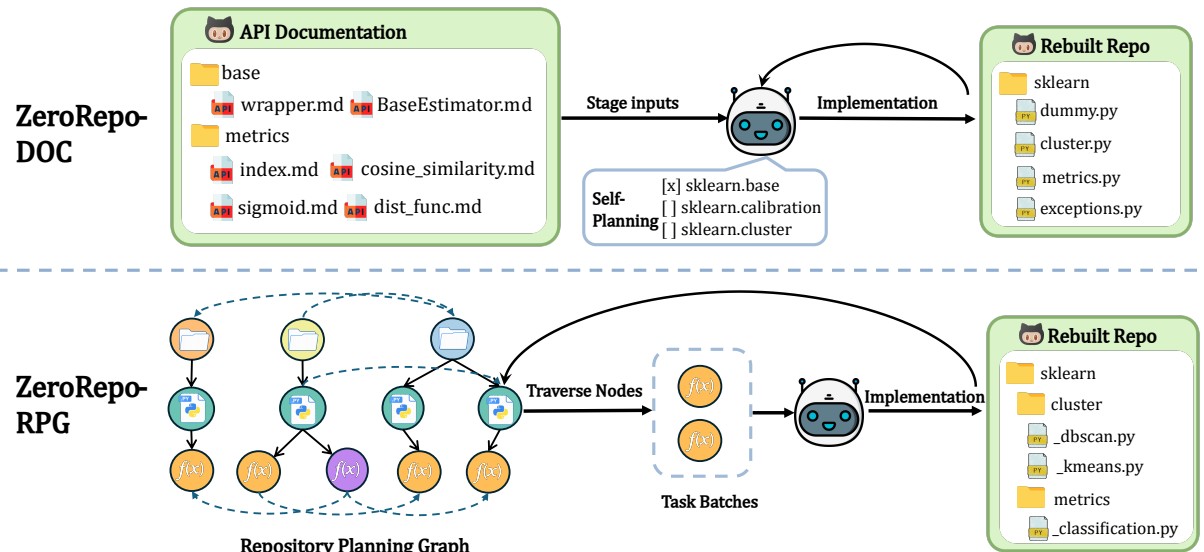

*Figure 11.* Comparison of reconstruction workflows adapted from ZeroRepo. **Top (ZeroRepo-Doc):** The baseline relies on unstructured API documentation, requiring the agent to perform autonomous self-planning and manual state tracking. **Bottom (ZeroRepo-RPG):** Our method leverages the extracted RPG as a structured prior, enabling deterministic topological traversal and context-aware batched execution.

documentation generator, to compile the raw reStructuredText (reST) or Markdown files into a unified textual representation, as illustrated in Figure 10. This compiled documentation captures the official definitions of classes, functions, and module hierarchies, ensuring that the baseline agents operate on the exact same informational standard as human developers reading the official manual. As summarized in Table 7, the scale of this context is substantial: the compiled documentation spans a total of 7,320 files and over 2.5 million tokens across the six subject repositories, presenting a rigorous benchmark for unstructured long-context understanding.

### B.2.2. BASELINES

**Backbone Framework: ZeroRepo.** We adopt ZeroRepo (Luo et al., 2025), a state-of-the-art framework originally architected for zero-shot repository generation, as our unified execution backbone. In its native configuration, ZeroRepo operates by synthesizing a Repository Planning Graph (RPG) from high-level user intents. This graph serves as a rigid execution schedule, guiding the agent through a topological traversal where functionalities are implemented and verified via Test-Driven Development (TDD). The framework's modular design decouples *planning* (Graph Construction) from *execution* (Code Generation), making it an ideal substrate for isolating and comparing different planning strategies.

**Adaptation for Reconstruction.** For the task of Repository Reconstruction, the objective shifts from hallucinating new systems to recovering existing ground-truth architectures. To evaluate the efficacy of our extracted RPG against unstructured information, we adapt ZeroRepo into two distinct configurations by modifying its planning source:

**ZeroRepo-Doc (Unstructured Baseline).** In this configuration, we lobotomize the graph-based planning engine to simulate a conventional developer workflow. The agent is initialized solely with the raw API documentation of the target repository and an empty progress tracking file. Without a pre-computed graph, the reconstruction proceeds in an open-ended, iterative loop:

- **Self-Planning**: In each iteration, the agent must manually cross-reference its progress against the documentation to identify pending modules and formulate its own immediate objectives.

- **Manual State Tracking**: The agent bears the full cognitive burden of global state maintenance, creating and updating a checklist to prevent redundant or missed implementations.

- **Execution**: It employs standard TDD, writing reproduction scripts and code based on its self-determined plan. The

process terminates only when the agent subjectively judges that all documented requirements are met.

**ZeroRepo-RPG (Graph-Guided Reconstruction).** In our proposed configuration, we replace the generative planner with the **RPG-Encoder**. Instead of generating a graph from scratch, we inject the pre-extracted RPG as a ground-truth topological prior. This shifts the burden of planning from the agent to the substrate, transforming reconstruction into a deterministic traversal:

- **Topological Traversal**: The extracted nodes are arranged into a strict **dependency-based topological order**. This guarantees that prerequisite modules are implemented before dependent ones, eliminating circular dependency errors.

- **LLM-Driven Batching**: To optimize efficiency, an LLM scheduler previews the topological queue and dynamically aggregates semantically coherent nodes (e.g., a set of related utility functions) into a single implementation batch.

- **Context-Aware Execution**: The system executes these batches sequentially. The agent is relieved of global planning and focuses purely on implementation, leveraging the explicit context $(f, \mathbf{m})$ provided by the graph nodes to generate code that aligns with the established architecture.

### B.2.3. METRICS

To rigorously evaluate the fidelity of repository reconstruction, we adapt metrics from RepoCraft (Luo et al., 2025) to capture four complementary dimensions: *functional coverage*, *spectral novelty*, *executable fidelity*, and *structural scale*. These metrics collectively assess (i) whether the ground-truth functionalities are successfully recovered, (ii) whether the model introduces extraneous capabilities, (iii) whether the reconstructed logic is semantically correct, and (iv) the complexity level of the realized system.

**Functionality Coverage (Recovery Rate).** This metric quantifies the **recall** of the reconstruction process—specifically, what fraction of the ground-truth functional topology has been successfully restored. We define the reference feature set $\mathcal{C} = \{c_1, \ldots, c_K\}$ using the ground-truth repository's functional signature (derived from developer documentation or canonical feature lists). The generated functionalities $\mathcal{G} = \{g_1, \ldots, g_N\}$ are extracted from the reconstructed codebase. To measure alignment, we employ K-Means clustering with $\mathcal{C}$ as fixed centroids, mapping each generated feature $g_i$ to its nearest reference intent $f(g_i) \in \mathcal{C} \cup \{c_{\text{OOD}}\}$ (where $c_{\text{OOD}}$ denotes Out-Of-Distribution). Feature assignments are further refined by an LLM-as-Judge to mitigate semantic drift. Coverage is defined as the proportion of ground-truth categories "hit" by the reconstruction:

$$\text{Coverage} = \frac{1}{|\mathcal{C}|} \sum_{j=1}^{K} \mathbb{1}\left[\exists g_i \in \mathcal{G}, \ f(g_i) = c_j\right]. \tag{6}$$

A higher coverage indicates that the RPG successfully guided the agent to rebuild a more complete functional footprint of the original system.

**Functionality Novelty (Extrapolation).** In the context of reconstruction, pure imitation is not always possible; models may "hallucinate" valid but unrequested features. To capture this behavior, we measure the proportion of generated functionalities that fall outside the ground-truth taxonomy. Novelty is calculated as the fraction of generated nodes assigned to the $c_{\text{OOD}}$ centroid:

$$\text{Novelty} = \frac{1}{|\mathcal{G}|} \sum_{i=1}^{N} \mathbb{1}\left[f(g_i) = c_{\text{OOD}}\right]. \tag{7}$$

While high novelty is desirable in open-ended generation, for reconstruction tasks, it characterizes the model's tendency to extrapolate or deviate from the strict blueprint provided by the RPG.

**Functionality Accuracy (Executable Fidelity).** Coverage validates intent, but fidelity requires correctness. We evaluate the semantic equivalence of the reconstructed code against the original repository's behavior using adapted test suites. We report two statistics:

- **Voting Rate (Feature Presence)**: The fraction of tasks where the validation pipeline confirms that a plausible implementation of the target algorithm exists. This measures the system's ability to *instantiate* the planned logic.

- **Success Rate (Semantic Correctness)**: The fraction of tasks where the reconstructed code passes the unit tests. This measures the *executable validity* of the implementation.

**Code-Level Statistics (Structural Scale).** Finally, to ensure the reconstruction is not merely a skeletal prototype, we compare the scale of the generated codebase against realistic standards. Metrics are computed over filtered source files (excluding non-production artifacts like `tests` or `benchmarks`) to assess structural complexity:

- **File Count**: Reflects the modular granularity and file-level organization.

- **Normalized LOC**: Effective lines of code (excluding comments/blanks), proxying implementation volume.

- **Code Token Count**: Total lexical tokens, indicating the density and complexity of the synthesized logic.

### B.2.4. MODEL CONFIGURATION

To strictly separate structural reasoning from coding capability, we utilize distinct models for the planning and execution phases. We employ **GPT-4o** (OpenAI, 2024) as the extraction engine to analyze the source repository and generate the Repository Planning Graph (RPG), leveraging its strong reasoning abilities to ensure high-fidelity structural representation. For the downstream reconstruction tasks within the ZeroRepo framework, we evaluate two different backbone models: **GPT-5-mini** and **GPT-4.1**. This variation allows us to assess the robustness of our method across models with differing parameter scales.

## C. More Results

### C.1. Repository Reconstruction

*Table 8.* Main results on **Requests** reconstruction tasks in RepoCraft.

| Framework | Backbone | Coverage (%) ↑ | Pass Rate (%) ↑ | #Files ↑ | nLOC ↑ | Code Tokens ↑ |
|---|---|---|---|---|---|---|
| Gold Projects (Reference) | Human Developers | 100.0 | 93.0 / 93.0 | 17 | 2273 | 22,297 |
| ZeroRepo-Doc (Baseline) | GPT-4.1 | 54.6 | 13.2 / 32.1 | 7 | 968 | 8141 |
| | GPT-5-mini | 72.7 | 27.5 / 54.9 | 28 | 1,890 | 15,842 |
| **ZeroRepo-RPG (Ours)** | GPT-4.1 | 100.0 | 88.4 / 95.3 | 23 | 2901 | 24,734 |
| | GPT-5-mini | 100.0 | 88.4 / 95.3 | 20 | 3503 | 27,188 |

*Table 9.* Main results on **Scikit-Learn** reconstruction tasks in RepoCraft.

| Framework | Backbone | Coverage (%) ↑ | Pass Rate (%) ↑ | #Files ↑ | nLOC ↑ | Code Tokens ↑ |
|---|---|---|---|---|---|---|
| Gold Projects (Reference) | Human Developers | 100.0 | 99.5 / 100.0 | 185 | 65,927 | 592,187 |
| ZeroRepo-Doc (Baseline) | GPT-4.1 | 68.1 | 53.9 / 64.0 | 181 | 6,787 | 69,468 |
| | GPT-5-mini | 72.3 | 55.6 / 66.3 | 76 | 12,007 | 101,988 |
| **ZeroRepo-RPG (Ours)** | GPT-4.1 | 100.0 | 79.3 / 99.5 | 269 | 71,835 | 698,602 |
| | GPT-5-mini | 100.0 | 82.8 / 99.5 | 267 | 96,806 | 900,381 |

*Table 10.* Main results on **Sympy** reconstruction tasks in RepoCraft.

| Framework | Backbone | Coverage (%) ↑ | Pass Rate (%) ↑ | #Files ↑ | nLOC ↑ | Code Tokens ↑ |
|---|---|---|---|---|---|---|
| Gold Projects (Reference) | Human Developers | 100.0 | 97.8 / 100.0 | 699 | 218,924 | 943,873 |
| ZeroRepo-Doc (Baseline) | GPT-4.1 | 33.3 | 36.4 / 48.2 | 29 | 605 | 5347 |
| | GPT-5-mini | 66.7 | 52.5 / 64.7 | 135 | 10,454 | 99,460 |
| **ZeroRepo-RPG (Ours)** | GPT-4.1 | 91.7 | 63.6 / 89.6 | 220 | 59283 | 565680 |
| | GPT-5-mini | 95.8 | 81.4 / 96.6 | 288 | 80,534 | 686,951 |

*Table 11.* Main results on **Statsmodels** reconstruction tasks in RepoCraft.

| Framework | Backbone | Coverage (%) ↑ | Pass Rate (%) ↑ | #Files ↑ | nLOC ↑ | Code Tokens ↑ |
|---|---|---|---|---|---|---|
| Gold Projects (Reference) | Human Developers | 100.0 | 96.8 / 100.0 | 271 | 83,325 | 893,824 |
| ZeroRepo-Doc (Baseline) | GPT-4.1 | 66.7 | 71.7 / 75.9 | 87 | 5,235 | 55,688 |
| | GPT-5-mini | 75.0 | 57.8 / 79.5 | 163 | 25,472 | 274,983 |
| **ZeroRepo-RPG (Ours)** | GPT-4.1 | 96.6 | 89.8 / 98.9 | 303 | 66,045 | 709,424 |
| | GPT-5-mini | 97.7 | 91.4 / 100.0 | 297 | 92,618 | 942,411 |

*Table 12.* Main results on **Pandas** reconstruction tasks in RepoCraft.

| Framework | Backbone | Coverage (%) ↑ | Pass Rate (%) ↑ | #Files ↑ | nLOC ↑ | Code Tokens ↑ |
|---|---|---|---|---|---|---|
| Gold Projects (Reference) | Human Developers | 100.0 | 99.0 / 100.0 | 217 | 106,447 | 943,873 |
| ZeroRepo-Doc (Baseline) | GPT-4.1 | 81.8 | 63.4 / 82.5 | 726 | 14,770 | 127,326 |
| | GPT-5-mini | 75.0 | 56.2 / 81.4 | 180 | 14,610 | 125,957 |
| **ZeroRepo-RPG (Ours)** | GPT-4.1 | 97.4 | 89.5 / 92.8 | 146 | 21,657 | 201,484 |
| | GPT-5-mini | 97.4 | 80.1 / 97.4 | 115 | 30,066 | 249,094 |

## C.2. Agent Behavior

We calculated the tool-to-action ratios across all evaluation episodes to quantify the behavioral strategies of each model, as shown in Figure 12. The results indicate that the agents generally maintain a robust balance between keyword-based retrieval and structural navigation. While `SearchNode` serves as a foundational tool for locating initial entry points, the significant utilization of `ExploreRPG`—especially by high-performing models like GPT-4o—highlights its critical role in the reasoning loop. This distribution suggests that capable agents effectively recognize the value of `ExploreRPG`, strategically employing it to leverage topological connections and gain a holistic understanding of the codebase, rather than relying solely on local keyword matches.

## C.3. Error Analysis

To look beyond aggregate metrics and understand the behavioral divergence between RPG-enhanced agents and baselines, we conducted a manual qualitative analysis on a stratified sample of failure cases. Based on the taxonomy defined in Table 14, we observe distinct error patterns that highlight the structural advantages of RPG while exposing persistent challenges in agentic reasoning.

## C.4. Cost Analysis

Beyond effectiveness and behavioral patterns, we further analyze the computational cost of different agents to assess their practical efficiency. We report the average token consumption and monetary cost per instance across all evaluation episodes.

Overall, RPG-enhanced agents incur moderately higher costs than lightweight baselines due to more frequent structured exploration and reasoning steps. However, this increase is well-controlled and scales sublinearly with performance gains, indicating that improved task success is not achieved through excessive tool usage or redundant interactions. In particular, high-performing models demonstrate a favorable cost–performance trade-off by allocating additional budget primarily to informative exploration rather than repeated failed actions.

We note that for LOCAGENT with `o3-mini`, the reported cost can be unusually low, as the agent often terminates after only 1–2 rounds and may occasionally produce an answer without invoking any tool. This reflects early termination behavior rather than efficient full-loop reasoning, and is specific to the LOCAGENT–`o3-mini` pairing.

For completeness, we provide the full cost breakdown for all models, including prompt tokens, completion tokens, and estimated monetary cost, in Table 15.

*Table 13.* Main results on **Django** reconstruction tasks in RepoCraft.

| Framework | Backbone | Coverage (%) ↑ | Pass Rate (%) ↑ | #Files ↑ | nLOC ↑ | Code Tokens ↑ |
|---|---|---|---|---|---|---|
| Gold Projects (Reference) | Human Developers | 100.0 | 83.0/ 100.0 | 681 | 109,457 | 917,622 |
| ZeroRepo-Doc (Baseline) | GPT-4.1 | 83.3 | 61.3 / 77.5 | 228 | 8114 | 687,718 |
| | GPT-5-mini | 83.3 | 66.2 / 81.8 | 276 | 16,053 | 135,523 |
| **ZeroRepo-RPG (Ours)** | GPT-4.1 | 100.0 | 96.2 / 98.1 | 408 | 43,468 | 391,260 |
| | GPT-5-mini | 100.0 | 91.9 / 97.5 | 369 | 61,702 | 496,570 |

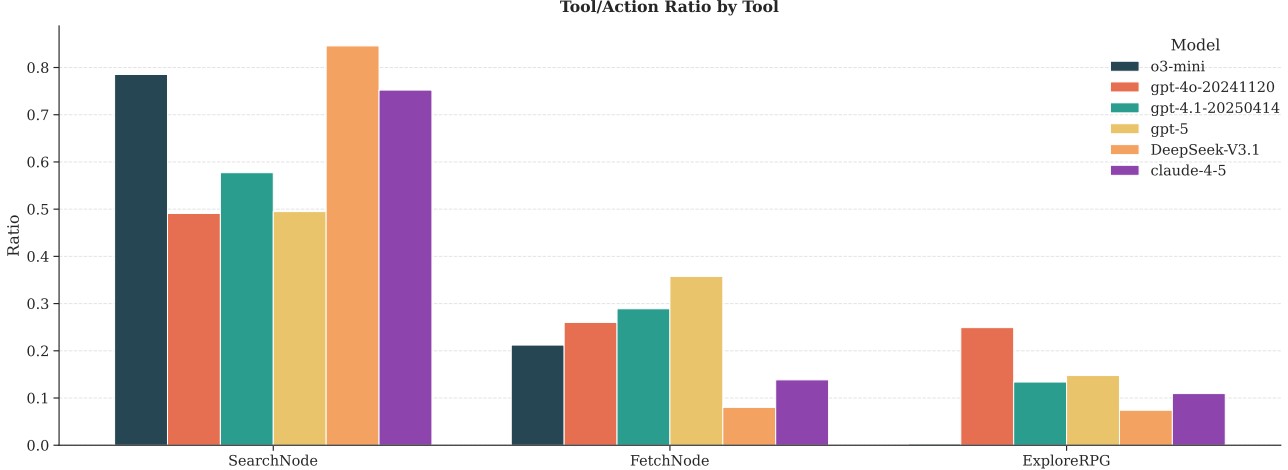

*Figure 12.* Breakdown of the Tool/Action ratio for each tool across different models. This metric reflects the frequency of specific tool invocations relative to the total actions performed.

# D. Ablation

## D.1. Repository Reconstruction

### D.1.1. ABLATION SETUP

To rigorously isolate the contribution of hierarchical topological signals to repository reconstruction, we conduct a controlled ablation study on the `scikit-learn` repository from the **RepoCraft** benchmark. All experiments utilize **GPT-5-mini** as the underlying reasoning engine. We define two specific ablation settings to evaluate the agent's capability in structural inference:

- **Function-Level Ablation (-Func):** In this setting, we strip fine-grained implementation guidance by removing metadata from leaf nodes (functions) and eliminating function-to-function dependency edges from the RPG. Consequently, the agent retains file-level boundaries but must autonomously deduce necessary function signatures from high-level features and independently derive their logical implementation order.

- **File-Level Ablation (-File/-Func):** Building upon the function-ablated graph, we further excise all file-level structural information, including file nodes and directory paths. This setting retains only the abstract semantic feature descriptions. It forces the agent to perform *ab initio* structural organization: the agent must semantically aggregate discrete features to synthesize file architectures, design class and function hierarchies, and plan the global implementation sequence without any reference directory topology.

### D.1.2. MORE RESULTS

**Analysis of Structural Clustering Behaviors.** Table 16 illustrates how different topological priors influence the LLM's organization strategies. (1) **Granular Encapsulation in Function Ablation.** When function-level metadata is removed, the model exhibits a preference for explicit object-oriented modeling. The significant increase in class count (524 → 815) and function definitions indicates that without specific procedural guidance, the agent tends to atomize logic into smaller, discrete

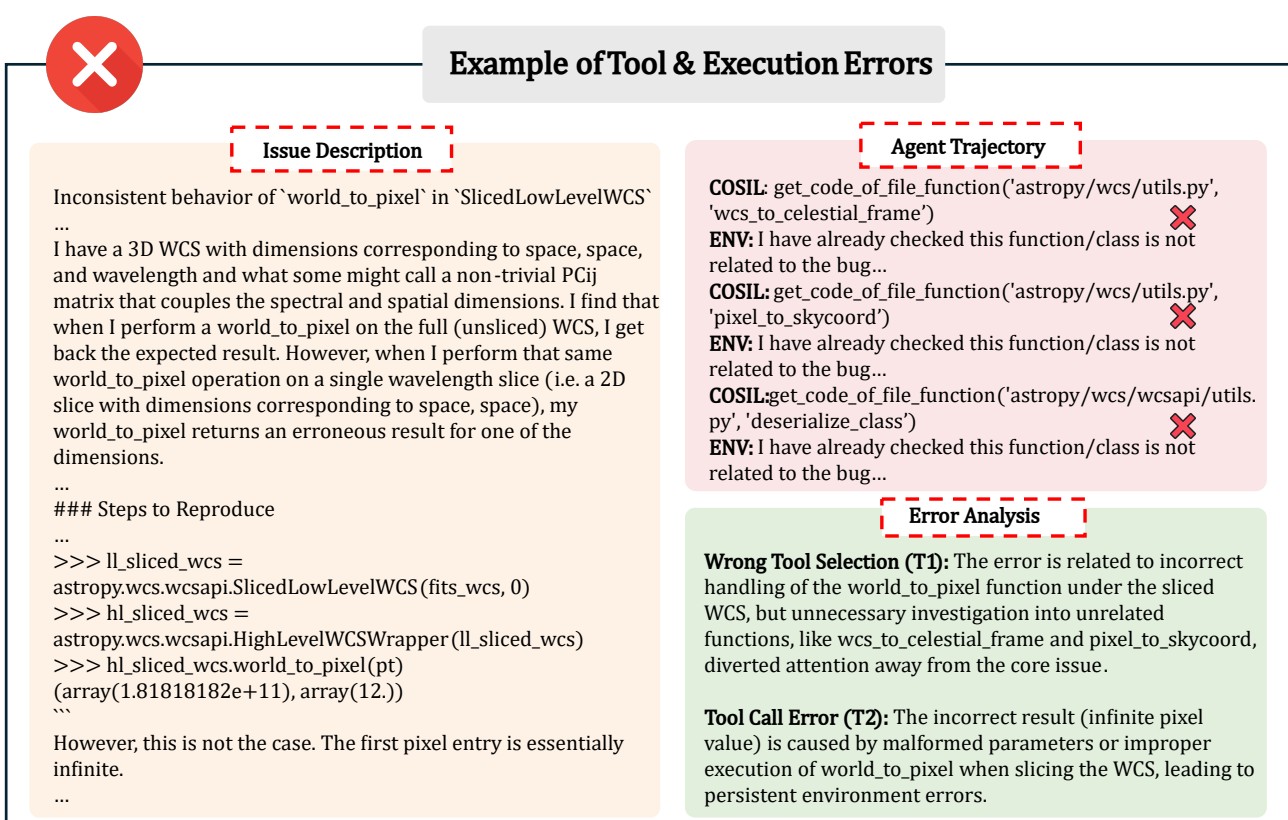

*Figure 13.* Case study of tool misuse and execution failure. COSIL exhibits **Wrong Tool Selection** by investigating unrelated functions (e.g., `wcs_to_celestial_frame`) instead of the core issue ('world_to_pixel'), leading to persistent environment errors and a failure to reproduce the bug.

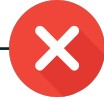

## Example of Search & Exploration Errors

### Issue Description

Mod function lambdify bug
Description:
When lambdifying any function of structure like `expr * Mod(a, b)` sympy moves the multiplier into the first argument of Mod, like `Mod(expr * a, b)`, WHEN we specify `modules=[]`

This is an example from Sympy online shell
```
>>> from sympy import Mod, lambdify, symbols
>>> x, y = symbols('x y')
>>> expr = -Mod(x, y)
>>> f = lambdify([x, y], expr)
>>> f(3, 7)
-3
>>> inspect.getsource(f)
def _lambdifygenerated(x, y):
    return (-mod(x, y))

>>> g = lambdify([x, y], expr, modules=[])
>>> g(3, 7)
4
>>> inspect.getsource(g)
def _lambdifygenerated(x, y):
    return (-x % y)
```

### Agent Trajectory

**Locagent**: search_code_snippet(search_terms ="lambdify","modules=[]"]})
**ENV:** sympy/utilities/lambdify.py:lambdify
sympy/integrals/risch.py:DifferentialExtension._repr_...
sympy/integrals/risch.py:Differ
...
**Locagent:**get_entity_contents(start_entities=["sympy/core/mod.py:Mod","sympy/utilities/lambdify.py:lambdify"])
**ENV:**
6 | class Mod(Function):
 7 |    """Represents a modulo operation on symbolic expressions.
8 |    Parameters
 ...
12 |    p : Expr
 13 |       Dividend.
**Locagent:**explore_tree_structure(start_entities="sympy/utilities/lambdify.py")
**ENV:** sympy/utilities/lambdify.py
── imports ── sympy/utilities/exceptions.py: SymPyDeprecationWarning
── imports ── sympy/core/compatibility.py:is_sequence
**Locagent  Final Answer:**
sympy/utilities/lambdify.py:function: lambdify: line: 177, sympy/core/mod.py:class: Mod:lines: 6-33

### Error Analysis

**Insufficient Coverage(T3):**The agent missed critical components related to code generation and printing, which directly lead to the bug. Key functions that deal with lambdify's code generation process and Mod's behavior were not explored, resulting in an incomplete error analysis.
**Query Formulation Error(T4):** The searches used by the agent were too broad and did not focus on the core issue. More targeted searches focusing on specific functions related to Mod and lambdify's internals would have yielded better results and identified the true cause of the issue.

*Figure 14.* Case study of a search and exploration failure. The agent attempts to resolve a `lambdify` bug in SymPy but fails due to **Insufficient Coverage (T3)**, missing critical code generation logic, and **Query Formulation Errors (T4)**, where searches were too broad to locate the root cause.

*Table 14.* Error Taxonomy for Code Localization Agents (T1–T12)

| Category | Error Type | Description |
|---|---|---|
| Tool & Execution Errors | Wrong Tool Selection (T1) | Agent uses an inappropriate tool for the task; e.g., using `grep` when `find` is needed, or reading files when search is more suitable. |
| | Tool Call Error (T2) | Malformed parameters, invalid paths/regex, or runtime failures (e.g., "No such file", syntax errors, timeouts). |
| Search & Exploration Errors | Insufficient Coverage (T3) | Agent fails to systematically explore relevant code areas; misses obvious entry points from the problem statement. |
| | Query Formulation Error (T4) | Searches use wrong keywords, overly broad/narrow scope, or miss obvious terms from the problem statement. |
| | Redundant Search (T5) | Agent makes repetitive tool calls; stuck in loops; repeats similar queries without progress. |
| Reasoning & Interpretation Errors | Evidence Misinterpretation (T6) | Agent misreads tool output; confuses similar symbols; incorrect understanding of code relationships. |
| | Info Synthesis Failure (T7) | Agent fails to connect evidence across multiple steps; doesn't build a coherent picture from findings. |
| | Hallucination (T8) | Agent invents non-existent files, functions, or paths; outputs fabricated information not grounded in evidence. |
| Context & Scope Errors | Problem Misunderstanding (T9) | Agent fundamentally misunderstands the task; focuses on the wrong aspect of the problem. |
| | Granularity Mismatch (T10) | Output at the wrong specificity level (too coarse/fine); e.g., file-level when function-level is needed. |
| | Codebase Context Failure (T11) | Agent fails to understand project structure, conventions, or architecture. |
| Multi-Factor Errors | Cascading Errors (T12) | Multiple interacting errors where early mistakes propagate; recovery attempts fail. |

class-based units for encapsulation. (2) **Feature Consolidation in File Ablation.** Conversely, the absence of file-level constraints shifts the behavior towards coarse-grained aggregation. The model consolidates features into fewer physical files ($250 \rightarrow 154$), resulting in a higher feature density per file ($19.77 \rightarrow 33.60$). This suggests that without directory boundaries to enforce separation, the model inherently clusters semantically related functionalities into larger, unified modules rather than distributing them across a file system hierarchy.

## D.2. Repository Understanding

### D.2.1. ABLATION SETUP

To quantify the individual contributions of structural connectivity and semantic annotation, we evaluate **ZeroRepo-RPG** under two degradation protocols:

- **Dependency Graph Ablation (w/o Dependency):** In this variant, we sever all static dependency edges ($\mathcal{E}_{\text{dep}}$) from the RPG. This ablation strictly limits the `ExploreRPG` tool by occluding execution logic; the agent loses the ability to traverse call graphs or trace upstream/downstream relationships, forcing it to navigate solely based on physical file hierarchy.

- **Semantic Feature Ablation (w/o Feature):** Here, we strip high-level feature descriptions ($f$) and functional subordination edges ($\mathcal{E}_{\text{feature}}$). This degradation fundamentally impairs semantic navigability: the `SearchNode` tool regresses from intent-based retrieval to rigid keyword matching, while `ExploreRPG` ceases to display functional hierarchies. Furthermore, retrieved nodes lack refined summaries, compelling the agent to infer utility solely from raw identifiers.

*Table 15.* Efficiency analysis comparing the average number of agent interaction rounds and cost per instance.

| LLM | Method | Steps | Cost ($) | Eff. |
|---|---|---|---|---|
| o3-mini | OrcaLoca | 10.13 | 0.10 | 4.86 |
| | LocAgent | **2.14** | **0.04** | **21.75** |
| | CoSIL | 18.95 | 0.18 | 4.07 |
| | **RPG-Encoder** | 6.53 | 0.11 | 7.07 |
| GPT-4o | OrcaLoca | 29.26 | 0.98 | 0.54 |
| | LocAgent | 8.95 | 0.50 | 1.35 |
| | CoSIL | 20.65 | 0.30 | 2.21 |
| | **RPG-Encoder** | **8.22** | **0.20** | **3.84** |
| GPT-4.1 | OrcaLoca | 20.22 | 0.46 | 1.48 |
| | LocAgent | 11.94 | 0.86 | 0.76 |
| | CoSIL | 19.77 | 0.24 | 3.10 |
| | **RPG-Encoder** | **6.75** | **0.18** | **4.63** |
| GPT-5 | OrcaLoca | 38.60 | 0.82 | 1.06 |
| | LocAgent | 6.48 | 0.49 | 1.64 |
| | CoSIL | 19.52 | 0.31 | 2.64 |
| | **RPG-Encoder** | **6.34** | **0.22** | **4.25** |
| Claude-4.5 | OrcaLoca | 11.32 | 0.88 | 0.91 |
| | LocAgent | **7.54** | 0.88 | 0.85 |
| | CoSIL | 12.60 | **0.48** | **1.64** |
| | **RPG-Encoder** | 16.75 | 1.32 | 0.71 |

*Table 16.* Feature Distribution across Different Modes

| Category | Full Mode | | | Func Ablation | | | File Ablation | | |
|---|---|---|---|---|---|---|---|---|---|
| | Count | Features | Avg±Std | Count | Features | Avg±Std | Count | Features | Avg±Std |
| File | 250 | 4943 | $19.77 \pm 20.94$ | 275 | 6664 | $24.23 \pm 27.11$ | 154 | 5175 | $33.60 \pm 36.99$ |
| Class | 524 | 3584 | $6.84 \pm 6.51$ | 815 | 3676 | $4.51 \pm 4.76$ | 466 | 2462 | $5.28 \pm 6.16$ |
| Function | 1366 | 1359 | $0.99 \pm 0.07$ | 1597 | 2988 | $1.87 \pm 2.07$ | 1202 | 2713 | $2.26 \pm 2.25$ |

### D.2.2. MORE RESULTS

**Efficiency Degradation from Component Loss.** Table 17 quantifies the operational overhead introduced by removing structural and semantic priors. The full **RPG-Encoder** model consistently achieves the minimal trajectory length and cost across both LLMs, validating the synergistic efficiency of the complete graph. (1) **Impact of Semantic Features:** Removing feature metadata (*w/o Features*) incurs the most significant penalty in exploration steps (e.g., rising from 8.22 to 9.23 on GPT-4o). Without high-level summaries to guide intent-based retrieval, the agent is forced into a trial-and-error loop, repeatedly fetching raw code to verify relevance, which drastically prolongs the search process. (2) **Impact of Dependency Structure:** Severing dependency edges (*w/o Dependency*) primarily inflates token cost (e.g., +$0.09 on GPT-4.1). Lacking explicit execution paths, the agent must manually traverse the file system and read broader contexts to deduce logical connections, leading to inefficient information consumption compared to the surgical navigation enabled by the full RPG.

*Table 17.* Impact of Dependency and Feature Ablations on Steps and Cost.

| Setting | GPT-4o | | GPT-4.1 | |
|---|---|---|---|---|
| | Avg Steps | Cost ($) | Avg Steps | Cost ($) |
| RPG-Encoder | **8.22** | **0.20** | **6.75** | **0.26** |
| w/o Dependency | 8.53 | 0.27 | 7.31 | 0.35 |
| w/o Features | 9.23 | 0.30 | 7.37 | 0.37 |

