# OpenReview forum: "Closing the Loop: Universal Repository Representation with RPG-Encoder"
_ICML.cc/2026/Conference — ICML 2026 regular_

### Official Review · Reviewer_Thd3 · 2026-03-11

**Soundness:** 3
**Presentation:** 3
**Significance:** 4
**Originality:** 3
**Overall Recommendation:** 4
**Confidence:** 4

**Summary:**

The authors focus on proposing RPG-Encoder, a framework that converts a software repository into a Repository Planning Graph (RPG). This graph representation captures the structural and semantic relationships between files, functions, and modules. Moreover this model allowing LLM agents to better understand and navigate large code repositories.

**Compliance With Llm Reviewing Policy:**

Affirmed.

**Key Questions For Authors:**

Could the authors clarify how nodes and edge types are defined in the RPG? more details would be helpful for readers to better understand the model.

**Limitations:**

Could the authors discuss the limitations of the proposed method?

**Strengths And Weaknesses:**

Strengths

- Introduces a graph-based representation for repository-level code understanding.
- Addresses an important challenge in large codebase navigation for LLM agents.
- Provides tools that allow agents to search, retrieve, and explore repository structures.

weakness:
- Although the paper provides sufficient information about the experiments but lack a clear discussion of the limitations.
- The computational cost of building the model (RPG) could be more highlighted.

---

> ### Author Rebuttal · Authors · 2026-03-30
>
> We sincerely thank you for the positive assessment and for recognizing the core value of our work. We address your specific points below.
>
> **W2: Computational cost of RPG building**
>
> We fully agree that highlighting the computational cost is crucial for evaluating the practical utility of our method. In fact, mitigating this exact overhead for large, frequently updated repositories was the primary motivation for our **Incremental Evolution** mechanism (Section 3.2). And we already discussed RPG extraction cost in Section 7.2.
>
> As shown in Fig. 3 and Table 6 (Lines 345–380), the initial extraction cost is **495K tokens**. Over a 28-commit sequence, incremental updates consume 633K tokens, which is 14.7M fewer than rebuilding from scratch (a 95.7% reduction), while preserving downstream retrieval quality (Func Acc@5 remains around 81.9%).
>
> In the revision, we will emphasize this tradeoff earlier and more explicitly, ensuring a clear distinction between the one-time initialization cost and the highly efficient, recurring maintenance cost.
>
>
> **Q1: Definition of nodes and edge types.**
>
> We agree that the current presentation can be made more explicit in the main text, and we will revise this part accordingly in the next version.
>
> **Macroscopic Overview:** At a high level, RPG is designed as a **dual-view hierarchical graph**. It represents a repository not merely as a flat list of files, but as an organized ecosystem. Low-level physical code entities (functions, classes) are grouped under high-level semantic concepts (modules, subsystems). The connections between them capture both *how they interact structurally* (execution/dependency) and *what they mean logically* (semantic/functional hierarchy).
>
> To provide a clearer breakdown, we summarize the formal definitions of nodes and edges in **Table 1**. In the revision, we will elevate a refined version of this summary into Section 3 so that readers can grasp the graph construction intuitively on a first pass.
>
> **Table 1. Summary of node and edge definitions in RPG**
> | Component | Name | Definition | Example |
> |---|---|---|---|
> | **Node Rep.** | $v=(f,m)$ | Each node combines a semantic feature $f$ with structural metadata $m$. | For `load_config()`, $f$ describes its role; $m$ records its type and path. |
> | Feature | $f$ | A concise functional description of the code entity. | "Loads and validates experiment configuration." |
> | Metadata | $m$ | Repository-grounded structural information (type, path, etc.). | `{type=function, path=utils/config.py}` |
> | **Node Set** | $V = V_H \cup V_L$| RPG contains both high-level semantic nodes and low-level entity nodes. | A graph includes both a "training pipeline" node and a `train.py` node. |
> | Low-level | $V_L$ | Atomic physical code entities (files, classes, functions). | `model.py`, `Trainer`, `load_config()` |
> | High-level | $V_H$ | Abstract functional groupings induced from semantics. | "Data preprocessing module", "Evaluation subsystem" |
> | **Edge Set** | $E = E_{\text{feat}} \cup E_{\text{dep}}$| RPG contains both semantic/functional edges and structural dependency edges.| A function belongs to a subsystem AND calls another function. |
> | Functional | $E_{\text{feat}}$ | Edges encoding the semantic or teleological hierarchy. | `build_dataloader()` $\rightarrow$ "Data preprocessing module" |
> | Dependency | $E_{\text{dep}}$ | Edges encoding implementation-level relations (imports, calls). | `train.py` imports `model.py`; `train_step()` calls `forward()` |
>
> **W1: Limitations & Future Work**
>
> We appreciate the suggestion to explicitly discuss the boundaries of our current study. In the revised manuscript, we will add a dedicated **Limitations & Future Work** section to concisely address the following:
> 1. **Language Generalization:** Our current framework utilizes Python-centric AST parsers. Extending RPG-Encoder to multi-language or polyglot repositories will require integrating broader parsing tools.
> 2. **Large-Scale Resource Platform:** We currently extract RPGs on-demand. A key future direction is scaling this process across open-source platforms (e.g., GitHub) to build a massive, pre-extracted RPG resource hub to support broader community research.
>
> Thank you again for your thoughtful and constructive feedback. We would be happy to answer any further questions during the upcoming discussion phase.

---

### Official Review · Reviewer_tYW3 · 2026-03-13

**Soundness:** 3
**Presentation:** 3
**Significance:** 2
**Originality:** 2
**Overall Recommendation:** 4
**Confidence:** 4

**Summary:**

This paper studies the problem of repository-level representation for large codebases. The authors investigate the question of whether a unified intermediate representation can bridge repository understanding and repository generation. The paper proposes RPG-Encoder, a framework that encodes an existing code repository into a Repository Planning Graph (RPG) that integrates semantic descriptions and dependency structures.

The RPG-Encoder extracts semantic features from code entities (e.g., functions, files, directories) and organizes them into a hierarchical graph structure with functional and dependency edges. The framework also introduces an incremental evolution mechanism that updates the graph using commit-level changes, which reduces maintenance cost. Furthermore, the RPG is used as a unified interface for LLM agents, providing operations such as node search, traversal, and retrieval.

The method is evaluated on two tasks: repository understanding (code localization on SWE-bench benchmarks) and repository reconstruction (rebuilding repositories using RepoCraft). Experimental results show that RPG-Encoder improves localization accuracy and reconstruction coverage compared with several baseline methods.

Overall, this study's main aspect is to explore RPG as a unified and evolvable representation for repository reasoning, aiming to connect semantic intent with code structure

**Compliance With Llm Reviewing Policy:**

Affirmed.

**Key Questions For Authors:**

1. The graph construction relies on LLMs to extract semantic features from code. How robust is this process in practice? Will the LLM generate hallucinated or incorrect semantic descriptions? How do the authors detect or mitigate such errors?

2. When the repository becomes large, the RPG may also grow significantly. How does the proposed method ensure efficient retrieval and reasoning over large graphs?

3. The paper briefly introduces Agentic Tools for interacting with RPG. Could the authors provide more details on how these tools are implemented and how agents utilize them during reasoning?

**Limitations:**

no limitation section in this paper.

**Strengths And Weaknesses:**

Significance:
- This paper addresses an important problem in repository-level representation, which is relevant to many downstream tasks such as issue resolving, bug localization, and code search.
- The proposed RPG-Encoder provides a new perspective to represent repositories by constructing a semantically enriched graph structure. Such a representation could potentially support repository-level reasoning and provide useful interfaces for LLM agents.
- In theory, a graph-based representation like RPG could encode richer semantic information of a repository and provide structured signals for downstream tasks.
- However, I would like to see more practical analysis of this approach. First, constructing and maintaining such graphs may introduce non-trivial computational cost, especially for large and frequently updated repositories. Second, the graph construction relies heavily on LLM-based semantic extraction. It is unclear how robust this process is. LLMs may generate hallucinated or inaccurate semantic descriptions. The paper does not provide sufficient discussion on how such errors may affect the RPG quality or how they can be mitigated. Third, when the repository graph becomes large and complex, retrieving relevant nodes may become difficult. The paper lacks discussion on the scalability and retrieval efficiency of the proposed representation in large-scale repositories.

Originality:
- The idea of using RPG for repository localization and reasoning tasks is interesting, and the experimental results appear promising.
- However, the Repository Planning Graph (RPG) itself was proposed in prior work. This paper mainly focuses on encoding existing repositories into RPG rather than introducing a fundamentally new representation.
- As a result, the overall novelty of the paper is somewhat limited. The main contribution lies in the RPG encoding and maintenance framework, rather than the representation itself.

Presentation:
- The paper is generally well-written and easy to follow.
- However, more explanation of the RPG structure would help readers understand the framework more clearly. For example, the process of constructing the graph from raw code could be explained in more detail.
- The description of operations on RPG is also somewhat brief. In particular, the implementation details of the Agentic Tools (SearchNode, FetchNode, ExploreRPG) are not sufficiently clear. It would be helpful to include more concrete examples or case studies to illustrate how these tools interact with the RPG during repository reasoning.

---

> ### Author Rebuttal · Authors · 2026-03-30
>
> We sincerely thank you for constructive suggestions. We address your concerns below.
>
> **W1. Originality & Core Contribution**
>
> While RPG was introduced in prior work for repo generation, extending it to existing, large, and continuously evolving repositories is fundamentally different and non-trivial. Our contribution is therefore not the graph abstraction alone, but closing the repository-level reasoning loop by:
> 1. **Elevating RPG into a bidirectional substrate**: We extend RPG from a static planning scaffold into a unified intermediate representation that supports *both* repository comprehension and reconstruction.
> 2. **Enabling repository-scale use**: We introduce semantic lifting, hierarchical reorganization, and incremental evolution to make this practical at repository scale. Strong results on SWE-bench and RepoCraft validate the value of this extension in practice.
>
> **W2. Practicality: Computational & Maintenance Cost**
>
> This cost concern was in fact the primary motivation for our Incremental Evolution mechanism, and we have analyzed it in the paper (Section 7.2).
>
> As shown in Fig. 3 and Table 6 (Lines 345–380), the initial extraction cost is **495K tokens**. Over a 28-commit sequence, incremental updates consume 633K tokens, which is 14.7M fewer than rebuilding from scratch (95.7% reduction), while preserving downstream retrieval quality (Func Acc@5 remains 81.9%).
>
> **W3. Robustness of RPG Constrcution**
>
> We mitigate inaccurate semantic extraction through 3 mechanisms:
> 1. **Task Decomposition**: Graph construction is decomposed into atomic sub-tasks (feature extraction, refactoring), which modern LLMs handle precisely.
> 2. **Structural Grounding**: Semantic nodes are strictly anchored to AST-extracted dependency edges. Hallucinated features lacking execution contexts are ignored.
> 3. **Downstream Validation**: 93.7% Acc@5 on SWE-bench and 98.5% coverage on RepoCraft prove extraction noise does not undermine utility.
>
> To explicitly quantify this, we conducted two new robustness experiments using different extraction LLMs:
>
> **(A) Human Evaluation of Extraction Quality:** We randomly sampled 100 instances and manually evaluated stage-wise extraction quality along 4 dimensions: Factuality, Hallucination Rate, Architecture Alignment, and Grouping Semantics. As shown in Tab 1, the extraction is consistently reliable, with low hallucination rates even for 8B LLMs.
>
> Tab 1: Semantic Lifting & Refactoring Quality (%) (↑ higher is better, ↓ lower is better)
> |Dimension|Metric|GPT-4o|Qwen3-32B|Qwen3-8B|
> |---|---|---|---|---|
> |Lifting|Factuality Performance ↑|98.8|91.0|88.3|
> ||Hallucination Rate ↓|1.4|1.7|2.5|
> |Refactoring|Architecture Alignment ↑|90.8|88.8|83.9|
> ||Grouping Semantics ↑|90.8|88.6|87.4|
>
> **(B) Sensitivity of Downstream Performance:** To assess whether extraction variation causes substantial downstream degradation, we evaluated RPGs extracted by GPT-4o and Qwen3-32B on SWE-bench Live using GPT-4.1 and GPT-5 as agent backbones. As shown in Tab 2, performance varies only slightly and remains well above all baselines, indicating that the framework is robust to the choice of extraction LLM.
>
> Tab 2: Downstream Sensitivity on SWE-bench Live
> |Agent Backbone|Extractor|File Acc@1|File Acc@5|Func Acc@1|Func Acc@5|
> |---|---|---:|---:|---:|---:|
> |GPT-4.1|GPT-4o|78.0|90.5|64.7|**81.9**|
> ||Qwen3-32B|**81.7**|**91.8**|**69.3**|81.3|
> |GPT-5|GPT-4o|**82.1**|**94.4**|**71.9**|**87.8**|
> ||Qwen3-32B|81.0|93.3|70.9|87.7|
>
> **Q2. Scalability & Retrieval Efficiency**
>
> Retrieval is not a bottleneck because we avoid dense embedding searches over the full graph or raw codebase.
>
> 1. **Lightweight & Iterative Retrieval:** `SearchNode` uses fast BM25 and fuzzy matching over compact semantic features, rather than heavy embedding-based retrieval, which keeps search efficient in practice. Moreover, retrieval operates in an iterative agent loop, where the LLM progressively narrows the search space instead of relying on a single perfectly precise step, making the framework naturally robust to occasional noisy or sub-optimal tool returns.
> 2. **Empirical Quality:** We evaluated 100 calls, using GPT-5.4 with manual verification to assess whether the retrieved results matched the LLM’s search intent. The hit rate was **70.7%** for code entities and **65.6%** for feature entities, suggesting that this lightweight retrieval is sufficiently accurate to effectively support the agent’s reasoning without becoming a scaling bottleneck.
>
> **Q3. Tool Example**
>
> We agree that the tool descriptions can be made more explicit in the main text. Currently, **Appendix A** provides detailed algorithms and examples for each stage of RPG extraction, while **Appendix A.3** specifies the inputs, outputs, and invocation examples of the agentic tools. Due to the character limit, we cannot include a full multi-step trajectory here, but we would be glad to provide a complete step-by-step execution trace during the next discussion phase if helpful.

---

> > ### Author Rebuttal · Reviewer_tYW3 · 2026-04-01
> >
> > Thanks for the clarification. I will raise my score to a positive score.

---

> > > ### Author Response · Authors · 2026-04-01
> > >
> > > Thank you very much for your kind acknowledgement and for taking the time to reconsider our rebuttal. We sincerely appreciate your positive update and are glad that our clarification addressed your concerns.

---

### Official Review · Reviewer_4yvf · 2026-03-13

**Soundness:** 3
**Presentation:** 3
**Significance:** 3
**Originality:** 3
**Overall Recommendation:** 5
**Confidence:** 4

**Summary:**

This paper proposed RPG-Encoder, which generalizes Repository Planning Graph (RPG) to a unified, bidirectional IR. The core design motivation is that code repository comprehension and generation are inver processes: generation expands intent into detailed code, while comprehension needs to compress implementation back to high-level intent. RPG-Encoder is evaluated against repo understanding (e.g. SWE-bench Verified, SWE-bench Live) and repo reconstruction (RepoCraft), exhibiting strong results on both tasks compared to baselines.

**Compliance With Llm Reviewing Policy:**

Affirmed.

**Final Justification:**

The rebuttal addressed my concerns, and I am increasing my evaluation accordingly.

**Key Questions For Authors:**

1. What is the costs of initial RPG construction?
2. How sensitive is this approach to the quality of the LLM used for RPG extraction?
3. How do one evlauate the quality of the extracted RPG itself? Are there evaluation metrics suhc as dependency edges correctness or precision of semantic feature extraction? Currently, the quality relies on evaluation of downstream tasks.

**Limitations:**

yes

**Strengths And Weaknesses:**

**Strengths**

- The design motivation is well-motivated, and treating repository comprehension and generation as inverse processes is conceptually appealing.
- **Extensive experiments.** A range of LLMs are tested (GPT families, DeepSeek-V3.1, Claude-4.5-sonnet) that includes very recent models on two benchmarks. Results show that Repo-Enc outperforms baselines with a substantial margins.
- **Efficiency.** The incremental evolution mechanism reduces cost by around 96% over full rebuilds while showing minimal degredation in downstream task performance.
- **Qualitative analysis:** good qualitative insights and error analysis in Section 7, showing how the proposed methods reduce navigational failures

**Weaknesses**

- **Originality and Novelty:** The idea of RPG originates from previous work (Luo et al., 2025). While it is a valid contribution to extend it to a unified encoding for comprehension, individual components such as dependency graph contruction and incremental graph maintenance are standard techniques in Software Engineering. The novelty primarily lies in their engineering integration.
- **Lack of failure analysis.** The paper could benefit from more failure mode analysis such as semantic lifting failure.
- **Repository Reconstruction Comparison is somewhat unfair.** The comparison ZeroRepo-Doc vs. ZeroRepo-RPG is somewhat unfair as the RQP provides explicit topological ordering and batching, while the Doc baseline must perform self-planning.

---

> ### Author Rebuttal · Authors · 2026-03-30
>
> We sincerely thank you for your comments. We address your concerns below.
>
> **W1. Originality & Novelty.**
>
> While RPG was introduced in prior work for repo generation, extending it to large, continuously evolving repositories is non-trivial. Our contribution is not the graph abstraction alone, but **closing the repository-level reasoning loop by**:
> 1. **Bidirectional Substrate**: We extend RPG from a static planning scaffold into a unified representation supporting both comprehension and reconstruction.
> 2. **Repository-Scale Use**: We introduce semantic lifting, hierarchical reorganization, and incremental evolution. Strong results on SWE-bench and RepoCraft validate this extension in practice.
>
> **W3. Fairness of repository reconstruction comparison.**
>
> We appreciate this concern and agree that the intent of this experiment should be stated more clearly.
>
> 1. **Documentation lacks explicit structural ordering, whereas RPG does.** We designed the ZeroRepo-Doc baseline to reflect the practical setting where agents must work with flat, unstructured documentation. Thus, the explicit ordering provided by RPG is not an unfair advantage, but a core strength of the representation.
> 2. **The advantage is not solely from dependency-order guidance.** In the ablation study (Table 4, L350–357, and Appendix D), we remove dependency edges, forcing the LLM to plan file/function realization order by itself, much like the documentation baseline. Even in this setting, the ablated RPG still achieves 91.5% coverage and a 90.9% pass rate, substantially outperforming ZeroRepo-Docs (72.3% coverage, 66.3% pass rate). This suggests that the advantage of RPG goes beyond topological ordering alone.
>
> **Q1. Cost of Initial RPG Construction**
>
> This cost concern was in fact the primary motivation for our Incremental Evolution mechanism, and we have already analyzed it in the paper (Section 7.2).
>
> As shown in Fig. 3 and Table 6 (Lines 345–380), the initial extraction cost is **495K tokens**. Over a 28-commit sequence, incremental updates consume 633K tokens, which is 14.7M fewer than rebuilding from scratch (95.7% reduction), while preserving downstream performance.
>
> **Q3. RPG Extraction Quality**
>
> We agree that the intrinsic quality of the extracted RPG should be evaluated directly:
>
> 1. **Dependency Edges:** extracted deterministically via rule-based Python AST and static analysis, independent of LLMs.
> 2. **Semantic Features:** to assess robustness, we extracted RPGs using GPT-4o, Qwen3-32B, and Qwen3-8B, and evaluated 100 instances using GPT-5.4 with human verification across four dimensions: Factuality, Hallucination, Architecture Alignment, and Grouping Semantics.
>
> As Tab 1 shows, extraction is highly reliable: factuality remains strong and hallucination rates stay very low across models.
>
> Tab 1: Semantic Lifting & Refactoring Quality (%) (↑ higher is better, ↓ lower is better)
> |Dimension|Metric|GPT-4o|Qwen3-32B|Qwen3-8B|
> |---|---|---|---|---|
> |Lifting|Factuality Performance ↑|98.8|91.0|88.3|
> ||Hallucination Rate ↓|1.4|1.7|2.5|
> |Refactoring|Architecture Alignment ↑|90.8|88.8|83.9|
> ||Grouping Semantics ↑|90.8|88.6|87.4|
>
> **W2. Failure Analysis:**
>
> Based on extraction trajectories across the evaluated models, we analyzed the main failure modes and will include them in the revision.
>
> For **Semantic Lifting**, the dominant errors are:
> 1. **Hallucinated or over-specified semantics:** the model adds unsupported methods or responsibilities (e.g., turning a simple config loader into a “distributed synchronization manager”).
> 2. **Shallow or weakly grounded descriptions:** the output is plausible but overly generic.
>
> For **Semantic Refactoring**, the main failure modes are:
> 1. **Semantically mismatched grouping:** features are assigned to buckets that do not align with the repository’s actual architecture (e.g., placing a logging utility under “Database Management”).
> 2. **Duplicate assignment or scope drift:** unrelated features are grouped together, or the same feature is assigned repeatedly.
>
> **Q2. Downstream Sensitivity**
>
> To test sensitivity to the extraction model, we evaluated RPGs extracted by GPT-4o and the open-source Qwen3-32B on SWE-bench Live using GPT-4.1/GPT-5.
>
> As Tab 2 shows, downstream performance varies only slightly across extraction backbones. Notably, using the open-source Qwen3-32B for extraction does not lead to noticeable degradation, indicating that **RPG-Encoder is robust to the choice of extraction LLM.**
>
> Tab 2: Downstream Sensitivity on SWE-bench Live
> |Encoder Model|Extraction Model|File Acc@1|File Acc@5|File Prec|File Rec|Func Acc@1|Func Acc@5|Func Prec|Func Rec|
> |---|---|---:|---:|---:|---:|---:|---:|---:|---:|
> |GPT-4.1|GPT-4o|78.0|90.5|81.4|69.0|64.7|**81.9**|72.1|52.6|
> ||Qwen3-32B|**81.7**|**91.8**|**85.3**|**72.6**|**69.3**|81.3|**73.7**|**55.9**|
> |GPT-5|GPT-4o|**82.1**|**94.4**|**85.4**|**76.2**|**71.9**|**87.8**|**78.1**|**61.1**|
> ||Qwen3-32B|81.0|93.3|84.7|75.5|70.9|87.7|77.5|61.0|

---

> > ### Author Rebuttal · Reviewer_4yvf · 2026-04-04
> >
> > Thanks for the responses, and I am increasing the confidence score and originality score.

---

### Official Review · Reviewer_cppS · 2026-03-13

**Soundness:** 4
**Presentation:** 3
**Significance:** 3
**Originality:** 3
**Overall Recommendation:** 4
**Confidence:** 3

**Summary:**

This paper proposes RPG-Encoder, which is a unified representation for reasoning over large code repositories. Specifically, RPG-Encoder combines semantic descriptions of code entities with dependency relationships, which enables agents to navigate repositories more effectively and maintain the graph incrementally as the code evolves. Experiments on SWE-bench Verified/Live and RepoCraft show that RPG-Encoder improves code localization and repository reconstruction performance, achieving higher localization accuracy and reconstruction coverage.

**Compliance With Llm Reviewing Policy:**

Affirmed.

**Final Justification:**

The author's response has resolved my questions.

**Key Questions For Authors:**

- Why is the final repair performance of RPG-Encoder on SWE-Bench Verified/Live, apart from localization tasks?
- Can open-source LLMs also perform well when using the tools provided by RPG-Encoder to finish the task?

**Limitations:**

yes

**Strengths And Weaknesses:**

Strength:
- This paper explores an important research direction for improving SWE agents’ capabilities of exploring and understanding repositories.
- The evaluation is relatively comprehensive, comparing RPG-Encoder against diverse baselines and involving different LLM backends.

Weakness:
- There’s a lack of baselines that do not include any structural priors, such as mini-SWE-agent and OpenHands. It is interesting to study whether most advanced LLMs nowadays can explore and understand the repository accurately without any structural priors.
- While the paper claims that RPG Evolution “effectively preserves the repository’s semantic integrity with negligible degradation”, Table 6 shows that it fails to maintain Acc@1 effectively, indicating that it can still lead to non-negligible degradation for some dimension.

---

> ### Author Rebuttal · Authors · 2026-03-30
>
> We sincerely thank you for the constructive feedback. We address your questions below.
>
> **W1. Baselines without Structural Priors**
>
> We agree that comparing against advanced general SWE agents (OpenHands, SWE-agent) without explicit structural priors is a crucial baseline. To address this, we evaluate these agents on repository localization tasks.
>
> As shown in Tab 1, while advanced LLMs can indeed navigate repositories to some extent natively, RPG-Encoder provides substantial additional value. By systematically organizing the search space, **our method achieves significantly higher localization accuracy across both benchmarks.**
>
> Tab 1. Localization performance of general SWE agents on SWE-bench
> |Benchmark|Model|Method|File Acc@1|File Acc@5|File Prec|File Rec|Func Acc@1|Func Acc@5|Func Prec|Func Rec|
> |---|---|---|---:|---:|---:|---:|---:|---:|---:|---:|
> |Verified|GPT-4.1|OpenHands|81.6|83.0|81.4|76.7|64.4|73.4|64.8|57.1|
> |||SWE-agent|80.6|84.4|79.8|78.5|64.8|75.0|65.7|59.8|
> |||**RPG-Encoder**|**82.6**|**93.2**|**83.6**|**79.3**|**68.7**|**83.4**|**71.0**|**62.4**|
> ||GPT-5|OpenHands|81.8|87.2|82.2|81.9|79.5|82.5|81.0|67.2|
> |||SWE-agent|86.0|89.6|84.5|83.7|81.5|85.7|83.5|68.0|
> |||**RPG-Encoder**|**91.9**|**97.7**|**91.1**|**89.1**|**83.4**|**93.6**|**84.5**|**76.9**|
> |Live|GPT-4.1|OpenHands|74.8|77.1|73.9|60.5|60.7|66.7|60.6|45.1|
> |||SWE-agent|71.5|77.6|69.2|61.1|57.8|70.2|58.4|49.9|
> |||**RPG-Encoder**|**78.0**|**90.5**|**81.4**|**69.0**|**64.7**|**81.9**|**72.1**|**52.6**|
> ||GPT-5|OpenHands|74.1|80.7|71.7|72.2|64.1|81.2|71.6|65.0|
> |||SWE-agent|76.6|83.2|71.2|73.1|61.7|80.0|68.0|62.8|
> |||**RPG-Encoder**|**82.1**|**94.4**|**85.4**|**76.2**|**71.9**|**87.8**|**78.1**|**68.5**|
>
> **W2. Wording of "Negligible Degradation" for RPG Evolution**
>
> We appreciate this precise observation. The reviewer is correct that the term "negligible" oversimplifies the impact on strict Acc@1 exactness. We will revise the wording to accurately reflect this as a **highly favorable cost-fidelity trade-off** rather than zero degradation.
>
> However, looking closely at the data in Table 6, this trade-off is exceptionally practical for maintaining evolving repositories:
> * **(1) Massive Scalability via 95.7% Cost Reduction:** Our incremental strategy reduces token consumption from 14.7M to merely 633K. Confining heavy computation to a one-time initialization is exactly what makes long-term repository maintenance practically feasible.
> * **(2) Nuanced Shifts rather than Strict Degradation:** While top-1 exactness takes a slight hit (GPT-4.1 Function Acc@1 drops from 67.4% to 64.7%), the broader retrieval contexts required by agents are effectively preserved or even marginally improved. Crucially, GPT-4.1's Function Acc@5 actually increases from 80.3% to 81.9%, andFile Acc@5 rises from 88.2% to 90.5%.
>
> **Q1. Final Repair Performance**
>
> Yes. To answer your question regarding final repair capability beyond localization, we integrated RPG-Encoder into a strong repair pipeline and evaluated it end to end against the strongest baselines. **Concretely, we plug RPG-Encoder into Trae-agent as its repository reasoning substrate.** As shown in Tab 2, this integration improves final repair performance, indicating that the benefit of RPG-Encoder extends beyond search to full issue resolution.
>
> Tab 2: Final repair performance (Pass@1)
> |Model|Agent|SWE-bench Verified|SWE-bench Live|
> |---|---|---:|---:|
> |GPT-4.1|SWE-agent|46.8|15.6|
> ||Trae-agent|48.2|16.1|
> ||OpenHands|43.2|16.6|
> ||**RPG-Encoder**|**53.0**|**20.0**|
> |GPT-5|SWE-agent|64.1|25.3|
> ||Trae-agent|68.0|25.2|
> ||OpenHands|64.4|25.6|
> ||**RPG-Encoder**|**71.8**|**28.3**|
>
> **Q2. Can open-source LLMs perform well with RPG tools?**
>
> We further conduct experiments with open-source backbones. As shown in Tab 3, pairing models such as Qwen3-8B and Qwen3-32B with RPG-Encoder yields strong localization performance and consistently outperforms competing other methods. This suggests that the benefit of RPG transfers effectively to open-source LLMs.
>
> Tab 3. Localization performance with open-source LLMs
> |Model|Method|File Acc@1|File Acc@5|File Prec|File Rec|Func Acc@1|Func Acc@5|Func Prec|Func Rec|
> |---|---|---:|---:|---:|---:|---:|---:|---:|---:|
> |Qwen3-8B|LocAgent|52.9|62.1|56.6|45.8|33.9|38.2|36.6|22.7|
> ||CoSIL|51.2|69.0|56.0|44.5|32.4|43.0|42.0|26.3|
> ||OrcaLoca|56.5|67.1|63.1|50.8|44.1|48.0|46.7|31.3|
> ||**RPG-Encoder**|**59.9**|**69.1**|**64.3**|**51.6**|**44.5**|**53.7**|**50.7**|**32.6**|
> |Qwen3-32B|LocAgent|63.2|75.0|66.6|53.7|46.0|56.8|53.8|42.1|
> ||CoSIL|53.7|69.5|57.9|46.3|32.2|55.4|43.2|25.6|
> ||OrcaLoca|59.5|70.5|65.8|52.4|45.2|51.2|49.9|37.6|
> ||**RPG-Encoder**|**63.2**|**77.9**|**67.4**|**54.9**|**48.9**|**63.2**|**57.2**|**48.7**|
> |Kimi-K2.5|LocAgent|74.3|89.3|77.0|68.0|54.3|68.6|60.2|44.0|
> ||CoSIL|65.1|87.5|69.5|59.0|48.8|70.8|56.8|41.8|
> ||OrcaLoca|80.4|85.6|82.3|67.0|63.4|71.9|67.3|48.0|
> ||**RPG-Encoder**|**80.9**|**93.0**|**84.6**|**75.3**|**71.0**|**89.0**|**78.5**|**61.2**|

---

> > ### Author Rebuttal · Reviewer_cppS · 2026-04-04
> >
> > Thank you for the response. It has resolved my questions.

---

### Decision · Program_Chairs · 2026-04-30

**Decision:**

Accept (regular)

**Comment:**

This paper proposes RPG-Encoder, a framework that generalises the Repository Planning Graph (RPG) into a unified, evolvable representation for repository-level reasoning. The core idea is to treat repository comprehension and generation as inverse processes, closing the reasoning loop through semantic encoding of code entities, incremental graph evolution, and a structured agentic interface for LLM-based navigation.

All four reviewers recommend acceptance. Reviewers consistently highlighted the practical significance of the problem, the conceptual appeal of the comprehension-generation duality, and the strength of the experimental results across multiple benchmarks and LLM backends. The incremental evolution mechanism, which reduces maintenance overhead by over 95% with minimal performance degradation, was also noted as a practically valuable contribution.

Some concerns are worth noting for the revision. The RPG representation is best characterised as an engineering integration of established techniques rather than a fundamentally new representation. Reviewers also noted the lack of baselines without structural priors, a potentially asymmetric reconstruction comparison, and insufficient discussion of limitations, including robustness of LLM-based semantic extraction, scalability, and graph construction costs.

Overall, this is a solid contribution to an important and timely problem, and I recommend acceptance with the expectation that the authors address the above points in the final version.